# Cartographing dynamic stall with machine learning

Matthew Lennie[1], Johannes Steenbuck[1], Bernd R. Noack[2], and Christian Oliver Paschereit[1]

[1]Technische Universität Berlin, Institut für Strömungsmechanik und Technische Akustik
[2]LIMSI, CNRS, Université Paris-Saclay, Bât 507, rue du Belvédère, Campus Universitaire, F-91403 Orsay, France

**Correspondence:** Matthew Lennie (matthew.lennie@tu-berlin.de)

**Abstract.** Once stall has set in, lift collapses, drag increases and then both of these forces will fluctuate strongly. The result is higher fatigue loads and lower energy yield. In dynamic stall, separation first develops from the trailing edge up the leading edge, eventually the shear layer rolls up and then a coherent vortex forms and then sheds downstream with it's low pressure core causing a lift overshoot and moment drop. When 50+ experimental cycles of lift or pressure values are averaged, this
process appears clear and coherent in flow visualizations. Unfortunately, stall is not one clean process, but a broad collection of processes. This means that the analysis of separated flows should be able to detect outliers and analyze cycle to cycle variations. Modern data science/machine learning can be used to treat separated flows. In this study, a clustering method based on dynamic time warping is used to find different shedding behaviors. This method captures that secondary and tertiary vorticity vary strongly and in static stall with surging flow; the flow can occasionally reattach. A convolutional neural network was used
to extract dynamic stall vorticity convection speeds and phases from pressure data. Finally, bootstrapping was used to provide best practices regarding the number of experimental repetitions required to ensure experimental convergence.

**Keywords.** Wind Energy, Machine Learning, Unsteady Aerodynamics, Dynamic Time Warping, Clustering, Deep Learning, Data Science, Multi-Dimensional Scaling, Dynamic Stall, Dynamic Stall Vortex

# 1  Introduction

Beyond small angles of attack, airfoil boundary layers have to contend with strong adverse pressure gradients. When the boundary layer doesn't have enough momentum, a flow reversal occurs and eventually the flow separates from the surface of the airfoil (Abbott and Doenhoff, 1959). Once this occurs, viscous effects dominate and any assumption of potential flow falls apart (Schlichting and Gersten, 2016). This means that modeling separated flows has always been a challenging part of designing wind turbines or even understanding experimental and field data. Even in the age of Computational Fluid Dynamics

(CFD), attempts to simulate stall with Unsteady Reynolds Averaged Navier-Stokes (URANS) have not yet yielded good quality results (Strangfeld et al., 2015; Rumsey, 2008; Rumsey and Nishino, 2011). Large Eddy Simulations (LES) show promise but are still too computationally expensive to be used as an ordinary design and analysis tool (Rumsey and Nishino, 2011). In the wind industry, semi-empirical models (Andersen et al., 2007; Wendler et al., 2016; Holierhoek et al., 2013) are still the main analysis tools for stalled airfoil flows. These models have to make simplifications to be viable in terms of available

computational power and input boundary conditions, the key questions are: What information is lost? and if we had better understanding and better models, how much could we improve wind turbine designs?

Stall is the term used to describe a broad range of phenomena that occur during boundary layer separation. There are two broad characteristics that help us provide a loose definition;

1. A flow reversal in the boundary layer resulting in the stream-wise streamline no longer following the surface of the airfoil

(Abbott and Doenhoff, 1959)'. The region of flow reversal will usually have a neutral pressure.

2. The presence of instabilities, such as shear layer instabilities or wake mode (vortex shedding) instabilities (Hudy and Naguib, 2007). These instabilities make the pressure footprint on the airfoil highly unsteady.

While the following explanations of the categories of stall will dive deep into details, these two features remain the basic underlying phenomena.

Let us begin by considering a stationary airfoil. As the angle of attack increases, the airfoil will encounter trailing edge (light) stall (McCroskey, 1981). Light stall will develop at moderate angles of attack and is more likely to be present on airfoils with a well rounded leading edge (Greenblatt and Wygnanski, 2002; Leishman, 2006). The adverse pressure gradient overcomes the momentum of the boundary layer somewhere downstream of the point of minimum pressure (Abbott and Doenhoff, 1959). The vertical size of the viscous region will be in the order of the airfoil thickness (McCroskey, 1982). A well rounded leading

edge will result in a smooth development of trailing edge stall, whereas a sharp leading edge may cause trailing edge stall to be by-passed rapidly (Leishman, 2006). The separated region won't contribute to the lift implying a smooth roll off of the lift, increase of drag, and a nose up moment. Even on a stationary airfoil, the boundaries of the separated region will be unsteady (Mulleners and Rütten, 2016) and will vary along the span and chord.

At higher angles of attack, deep stall will develop on the airfoil (McCroskey, 1982). Deep stall is characterized by separation

occurring at the leading edge region. As the angle of attack increases, the point of minimum pressure will move closer to the leading edge as the stagnation point moves more towards the pressure side of the airfoil (Abbott and Doenhoff, 1959). Here the

airfoil leading edge geometry is critical as a tight radius will cause a stronger adverse pressure gradient which can lead to deep stall initiating from the leading edge thus by-passing light stall. Even though the stall occurs at the leading edge, the definition of "leading edge stall" usually involves a laminar bubble bursting but the mechanism can more simply be trailing edge stall that engulfs the entire suction side of the airfoil (Leishman, 2006). In the steady case, deep stall will cause a plummet in the lift being produced and a sharp increase in drag. The vertical size of the viscous region will be in the order of the airfoil chord (McCroskey, 1982). The viscous region will be home to various instabilities such as shear layer mode or wake mode shedding (Hudy and Naguib, 2007); essentially different types of shedding phenomena leading to fluctuating airfoil forces.

Flow that detaches from the leading edge can reattach due to transition of the shear layer (Abbott and Doenhoff, 1959) or a re-thickening of the airfoil e.g. wind turbine airfoils can have dents due to manufacturing (Madsen et al., 2019). This phenomenon is call a separation bubble. Bubbles are a sensitive phenomena and small changes to boundary conditions can make them disappear completely (Ward, 1963). Inflow turbulence, leading edge surface erosion, fouling or ice will often cause forced transition (Pires et al., 2018). Earlier transition will tend to reduce or remove bubbles (Ward, 1963). Even without outside influences, bubbles are an unstable phenomena due to shear-layer disturbances which lead to transition and eventual reattachment or bursting (Kirk and Yarusevych, 2017). For certain older airfoil families i.e. NACA 63-2nn, the presence or lack of a bubble may cause an airfoil to switch between leading and trailing edge stall this phenomenon is known as double stall (Bak et al., 1998). While double stall might no longer be as relevant on new generations of airfoils on wind turbines with pitch regulation, bubbles can affect stall behavior and the eventual performance of the airfoil.

What happens when the airfoil starts moving? When an airfoil moves from from low angles of attack into light stall regimes, there will be a phase lag between the angle of attack and the separation. This effect becomes stronger as the airfoil pitches faster and can be seen as a resistance to stall when compared to the stationary case. One can interpret this effect in a few ways:

1. The wake hasn't yet forgotten the previous flow arrangement, meaning the effective angle of attack is still catching up with the geometric angle of attack, i.e. circulatory lift delay.

2. The current boundary layer still has the higher momentum from the former more favorable flow state.

3. The surface of the airfoil accelerates the boundary layer during the motion.

When moving from light stall angles of attack down to attached flow, the flow attachment is delayed for the same reason. This appears in polar diagrams as hysteresis loops but can also be interpreted as a dangerous phase difference between the angle of attack and the lift, moment and drag. In this context, phase differences mean that the structure will absorb or dissipate energy (Bowles et al., 2014; Lennie et al., 2016). In short, this phase difference can lead to single degree of freedom pitch flutter also known as stall flutter (McCroskey, 1982). If the unstable nature of separated flows leads to the extent and phase of light stall to be variable between cycles of pitching, then it follows that the aeroelastic damping of the airfoil will also be variable between cycles [1].

---

[1]While we may normally consider an operating state to be stable or unstable on a long range trajectory, we may have to consider that each operating state can display short term behaviors that appear unstable

When an airfoil moves rapidly from attached flow into deep stall, it creates an effect known as dynamic stall. The separation moves rapidly from the trailing edge up to the leading edge, the shear layer becomes unstable and then rolls up into a vortex with a strong low pressure core (Mulleners and Raffel, 2013). The vortex then travels downstream causing a spike in low pressure across the airfoil which presents as a strong spike in lift and a strong dump in the moment. A full description of dynamic stall would be extraneous here but excellent reviews can be found in McAlister et al. (1978); McCroskey (1982, 1981); Leishman (2002); Carr (1987) and more modern experimental works can be found in Granlund et al. (2014); Mulleners and Raffel (2013); Mulleners et al. (2012); Mulleners and Raffel (2012); Muller-Vahl et al. (2017); Müller-Vahl et al. (2015); Strangfeld et al. (2015); Balduzzi et al. (2019); Holst et al. (2019)). For the discussion here, it is sufficient to note that as the strength and phase of the leading vortex varies, so will the aeroelastic stability.

To review the previous section:

1. There are different types of stall that occur differently in static or dynamic conditions.

2. The spatio-temporal variation is in both span and chord.

3. Differences in stall behavior will also lead to changes in aeroelastic stability.

So how are these variations treated? Treating stall as a stochastic process is a relatively recent idea. As early as 1978, one sees acknowledgment that stall is variable in literature such as McAlister et al. (1978); an experimental report that described it taking measurements of 50 cycles of a pitching airfoil undergoing dynamic stall to ensure convergence of the lift. While these researchers did acknowledge the variability of the data, they still used a simple average to represent the data. This was a reasonable choice at the time given that many of the more advanced tools now available did not exist nor was the requisite computational power available. Only more recently have researchers have begun to address the spatial and temporal variability of stall in experimental work. Mulleners and Raffel (2013) were able to show that dynamic stall could be described by two stages of a shear layer instability, and that the development of these instabilities varied across cycles. In light stall, it was shown that the trailing edge separation region had two modes, resulting in either a Von Karman shedding pattern or a stable dead water zone (Mulleners and Rütten, 2016). The separation pattern fluctuates unreliably and when vorticity is present, the vortex convection speed is also variable.

Experimental data from Manolesos serves as a detailed reminder that stall happens in 3 dimensions (Manolesos et al., 2014; Manolesos, 2014). Even on a simple 2D wind section, flow visualization showed four different separation patterns (Manolesos, 2014). These patterns are referred to as stall cells, and create complicated vortex patterns on and behind the airfoil. Even more complicated still are the separation patterns on wind turbine blades due to the changes of airfoil shape, twist and chord length (various surface visualizations can be found in Manolesos (2014); Lennie et al. (2018b); Vey et al. (2014)). Wind turbines uniquely experience very high angles of attack; where the spatial patterns complicate further (Skrzypinski et al., 2014; Skrzypinski, 2012; Gaunaa et al., 2016; Lennie et al., 2018b). The picture that should be now clear is that: stall is a continuum of behaviors rather than a small number of defined cases.

So variability is rampant in stall. How should we measure and interpret airfoil stall behavior? This paper will attempt to demonstrate that machine learning has provided a new set of tools that can be helpful for these very tasks. This paper will demonstrate:

1. A clustering method to group similar time series together.

2. A computer vision method for extracting vortex convection speeds from pressure data.

3. How to detect outliers and inspect the convergence of the dataset.

Furthermore, we will provide a future perspective on the way that machine learning may help us in modeling airfoil stall in simulations. While the specific methods used in this paper should prove to be useful and while we will point out some specific aerodynamic effects in the examples section, these are only examples. This paper is trying to communicate that machine learning more broadly is approachable and useful for unsteady aerodynamics, wind energy and other adjacent fields.

Before jumping into the new methods we should establish what kind of techniques have been used previously. It should be clear given the discussion so far that simple averaging or even phase averaging will remove important data (Riches et al., 2018). In dynamic stall, for example, phase averaged flow visualizations and pressure data appear vastly cleaner and more coherent than a single cycle. The cycle to cycle variations and outliers are an important part of the dataset and shouldn't be smeared out. Manolesos (2014) suggested conditional averaging to produce better airfoil polar diagrams. Mulleners and Rütten (2016) also

performed a kind of conditional averaging using the orbits of POD coordinates displayed onto recurrence plots. Furthermore, Holst et al. (2019) also suggests a binning approach, especially when considering very deep stall. Conditional averaging is an interesting approach, but the important question becomes, what rules should we use to split the data and is it possible to automate this process to some degree?

Fluid dynamics has always been a natural case for dimensionality reduction. In particular, there is abundant literature using

Singular Value Decomposition (SVD) based methods such as POD/PCA (Taira et al., 2017), DMD (Schmid, 2010; Kutz et al., 2015; Brunton et al., 2015), SPOD (Sieber et al., 2015). These methods generally do not perform well in cases with any kind of traveling wave behavior (Taira et al., 2017; Riches et al., 2018; Hosseini et al., 2016). The reason for this lies in the creation of fixed spatial functions/basis functions. If the shedding is consistent, the system will be sparse, a sensible reduced order system can be found. However, introduce phase jitter and the small number of basis functions no longer does a good

job in representing the shedding; so more mode shapes are needed. Even for a simple cylinder shedding, up to 50 modes were required to represent the system reasonably well (Loiseau et al., 2018). Dynamic stall convection velocities vary continuously (Mulleners and Rütten, 2016), therefore we cannot expect a sparse set of spatial functions to represent the system well.

Fortunately the SVD and simple averaging type methods are not the only forms of dimensionality reduction techniques available. It turns out the dimensionality reduction is a cornerstone technique of machine learning; an interactive summary

can be found in Christopher Olah's website (Olah, 2019). In this paper, we will show how Multi-Dimensional Scaling (MDS) (O'Connell et al., 1999) and clustering (Maimon and Rokach, 2006) can be used as a reliable analysis technique for airfoil stall. Nair et al. (2018) have demonstrated one approach to clustering for separated flows in the context of cluster-based feedback

control. Cao et al. (2014) also demonstrated the use of time series clustering in the context of combustion. The advantage of cluster type methods is that they break the data down into similar neighborhoods rather than assuming that a set of global basis functions can describe the whole domain. Both Loiseau et al. (2018) and Ehlert et al. (2019) have both demonstrated that Local Linear Embedding (LLE), a neighborhood type method, can create a sparse representation of the system. In this paper, we will focus on clustering and MDS, although other methods also show promise.

The MDS and clustering methods rely on a distance metric to gauge the similarity between the time series of lift of various experimental repetitions. As already discussed, the data will contain phase jitter which may cause simple distance metrics such as Euclidean to overestimate the difference between cycles (Ratanamahatana and Keogh, 2004). The problem is amplified by the strong gradients present around the time of vortex convection. This is a common time series problem and Dynamic Time Warping (DTW) was created for this purpose (Morel et al., 2018; Ratanamahatana and Keogh, 2004). DTW allows for the time series to be stretched and squashed a small amount to allow for an effective comparison between experimental repetitions. The approach of using a cycle to cycle distance metric (in this case DTW) is different to making time independent clusters used in the work of Nair et al. (2018). The difference in approach comes from intended application. In this paper, we will create clusters and MDS plots by comparing the entire time series of separate pitch cycles.

Methods such as clustering and MDS belong to a branch of machine learning called unsupervised learning, i.e. learning from the data without having the answer ahead of time. Conversely, supervised learning uses a labeled dataset to learn a mapping between input and outputs. Once a model is trained, we can then map new data. This is the nature of our second example, extracting the vortex convections from pressure data. We manually create a small set of examples by clicking on the vortex patterns, we then use this data to train a model that can do the same job over the whole dataset efficiently. Manually clicking on the patterns is a laborious, time wasting and unpleasant task. For these reasons, we want to do this only for the bare minimum number of examples. Fortunately, we can leverage the concept of transfer learning to minimize the effort.

The concept of transfer learning exploits the fact that once a model has been trained for one task, it can be easily re-molded to complete similar tasks (Brownlee, 2017). In practice this means that a neural network can be trained for a specific computer vision task and then easily be reused i.e. a network originally trained for classifying breeds of dogs within photographs can be easily reused on aerodynamics data (the FASTAI project has a lecture series expanding at length on this theme (Howard and Others, 2019)). This may seem like an exotic claim but there is solid reasoning underpinning the claim. Pictures are displayed in pixels which is an incredibly high dimensional space (modern cameras have in the 10 mega-pixel range). If we randomly choose pixel values, the chances of getting a sensible picture is almost zero, we would usually only get noise. This means that sensible pictures with geometric features such as lines and circles exists in an incredibly small neighborhood. That is to say, any real picture (of an elephant, a calculator, a cloud or even a plot of our pressure vs time) is more similar to any other real picture than it is to a picture of the kind of random static noise we know from old television sets. Why does this matter? It means that we can use any general picture dataset to get our neural network to the right neighborhood, that is, being able to recognize real geometry. It turns out that as far as the neural network is concerned, the pressure plots look close enough to real world pictures so that it only needs a small amount of re-training. Therefore, instead of requiring millions of training data examples, we only needed roughly 700.

In this paper, we will demonstrate the utility of transfer learning by using a pre-trained convolutional neural network (CNN) to extract vortex convection speeds from airfoil pressure plots. A huge challenge of working with experimental data is that it is exceptionally difficult to extract features from data in an automated fashion. One example of this is extracting the convection speed of a vortex from pressure data. To the human eye it is a fairly obvious stripe in the pressure plot, however it is challenging to extract this feature automatically based on basic rules. Computer vision machine learning is perfect for such cases. While the vortex convection speeds are themselves an interesting result, the example should demonstrate to readers the incredible power of using pre-trained neural networks for extracting features from data. Deep neural networks are becoming increasingly used within the wind industry for applications e.g. for predicting rotor icing (Yuan et al., 2019), power-curve estimation (Kulkarni et al., 2019) or even for rotor-blade inspections (Shihavuddin et al., 2019). We hope to demonstrate with this paper that modern machine learning tools and infrastructure can provide a useful boost to research in unsteady aerodynamics, wind energy and other adjacent fields.

## 2 Experimental Data

Most machine learning methods are heavily reliant on an initial dataset [2]. The analyses shown in the rest of this paper relies on two existing datasets. We use these two datasets to demonstrate different approaches as they feature different interesting effects. The wind tunnel dataset is the primary data source and unless explicitly stated will be in used in all figures, graphs and discussions. The towing tank dataset provides a great example for outlier detection. The following introductions aim to provide some context but do not exhaustively describe the experimental setups or the data they retrieved. The original references provide a far more detailed view into the setups.

### 2.1 Wind Tunnel

The first dataset was collected by Müller-Vahl (2015). Extensive unsteady aerodynamic experiments were conducted in a blowdown wind tunnel powered by a $75\,\text{kW}$ backward bladed radial blower. The test section is depicted in Figure 1 and is $610\,\text{mm}$ per $1004\,\text{mm}$. The model is mounted on two circular, rotatable plexiglas windows and the wind speed is measured with two hot-wire probes. The pressure around the model is captured by 20 pressure sensors on both suction- and pressure-side (40 in total). The NACA 0018 airfoil model has two control slots at 5% and 50% chord for additional blowing. The model has a chord length of $347\,\text{mm}$ and a span of $610\,\text{mm}$. More information about the tunnel can also be found in Greenblatt (2016) and excerpts of the dataset can be found at "https://www.flowcontrollab.com/data-resource".

The wind tunnel data covers a comprehensive collection of experiments with varying boundary conditions. The dataset has been thoroughly explored in previous publications and appears to be a good quality. It ranges from static baseline investigations over oscillating pitching and variation of free stream velocity (and a combination of both). In order to manipulate the boundary layer, blowing was added. One peculiarity of this data set is that boundary layer tripping can be induced by the taped over blowing slots on the suction side of the airfoil. For the purposes of our analysis, this detail was not critical.

---

[2]Most but not all. For example, Reinforcement Learning can use self-play as a training mechanism.

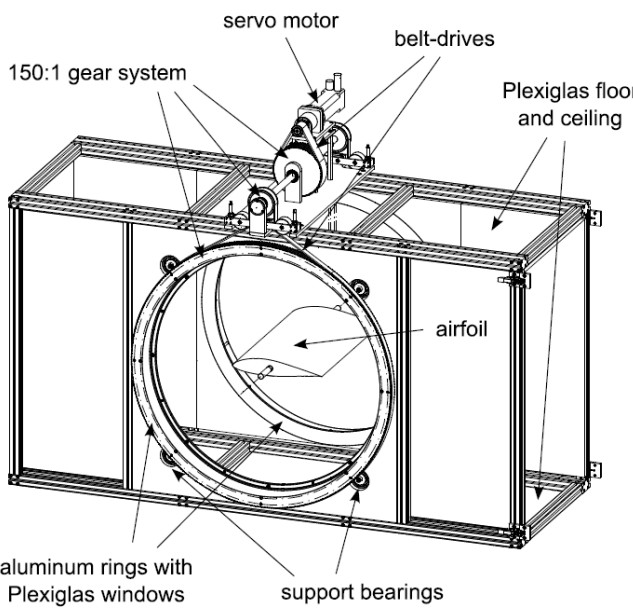

**Figure 1.** View of the test section showing the pitching mechanism and the approximate location of the airfoil model. From Müller-Vahl (2015).

## 2.2 Towing Tank

The second dataset comes from a large towing tank facility at the Technische Universität Berlin. This dataset is used to demonstrate outlier detection as the test configuration used in this data did have some peculiar stall behavior on some cycles. The water tank dimensions are $250\,\mathrm{m}$ length, a width of $8.1\,\mathrm{m}$, and an average depth of about $4.8\,\mathrm{m}$. A carriage runs on rails, towing a rig (and the model) through the water with a maximum speed of $12.5\,\mathrm{m\,s^{-1}}$. On it the complete measuring system is installed. The rig consists of to side plates with a length of $1.25\,\mathrm{m}$, a height of $1\,\mathrm{m}$, and a thickness of $0.035\,\mathrm{m}$ prohibiting lateral flow around the model. In between the side plates, the model, with a span size up to $1\,\mathrm{m}$ can be inserted at arbitrary angles of attacks. The model resembles a flat plate with an elliptical nose and blunt trailing edge. It has a span of $0.95\,\mathrm{m}$, $0.5\,\mathrm{m}$ chord a thickness of $0.03\,\mathrm{m}$. The surface is covered in aluminum and 12 pressure ports are inserted at the specified locations in Figure 2. The airfoil model is an unusual form but only some qualitative demonstrations are made with this dataset. A more detailed description is given in Jentzsch et al. (2019).

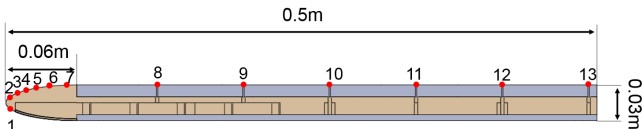

**Figure 2.** Cross section of the mounted flat plate. Red dots indicate position of pressure sensors. From Jentzsch et al. (2019).

## 3 Machine Learning Approaches

In this paper, we aim to provide a demonstration of a few machine learning methods and how they can be applied to unsteady aerodynamics data. A brief overview of the algorithms is provided to give a sense of what each of the algorithms is doing. The first algorithm demonstrates how to train and use a relatively simple machine learning algorithm, clustering, from scratch. The second examples demonstrates the more advanced deep learning approach and shows a few tricks to make it possible to do so with a modest amount of data and computational power. Usually each task will call for a different algorithm and different approach, but many of the principles discussed in the following section should transfer well onto other problems. This is especially true for the deep learning training tricks.

### 3.1 Dynamic time warping, clustering and multi-dimensional scaling

In this section, we will describe a method of grouping similar data together called clustering. For clustering to work we need two parts, a distance metric/measurement and a clustering algorithm. The distance metric gives us a measurement of similarity between our data[3]. The clustering algorithm takes the distances and groups the data into clusters.

Dynamic time warping (DTW) is a distance measurement that allows for squashing and stretching of the time series in order to reach a best fit. In practice, it's comparable to taking a winding path through a grid where each box corresponds to a time step from the two paths being compared (see Figure 5). The dynamic time warping algorithm is particularly useful in this case because it will still indicate that time series are similar even if there is a slight phase difference in vortex shedding or other stall phenomena (this effect is shown in the wave form in Figure 5)). The general rule of thumb is that a small amount of warping is a good thing, a lot can end up distorting reality. Therefore, DTW algorithms are usually implemented with either global or local constraints, these constraints have a bonus of increasing the computational efficiency.

A useful extension to the DTW algorithm creates a composite of multiple time series called a centroid (see Figure 3). Normally the problem with dynamic stall time series is that the vortex shedding is smeared out when simple means are taken. The onset of static stall can also appear to be a smooth process rather than a sudden separation that occurs at variable phases across different cycles of the experiments (see Figure 3). The barycenter extension to DTW creates an average that preserves these features. This means that the resulting centroid will be far more representative of a real stall process. In short, it is just an pseudo-average using different mathematics in the background, but it provides a better answer to the question: for these boundary conditions, what does the stall process typically look like?

For this research, the soft-DTW algorithm was used to compute the barycenter and was taken from the python module *tslearn* by Tavenard (2008). The algorithm was first proposed by Cuturi and Blondel (2017). To create the clusters, it is necessary to compare every time series within a group to each other. This means the complexity that the algorithm is $O(N^2)$. Two steps

---

[3]In this example, we are comparing a time series of a single experimental repetition against another. Clustering can also work with much more simple distance metrics

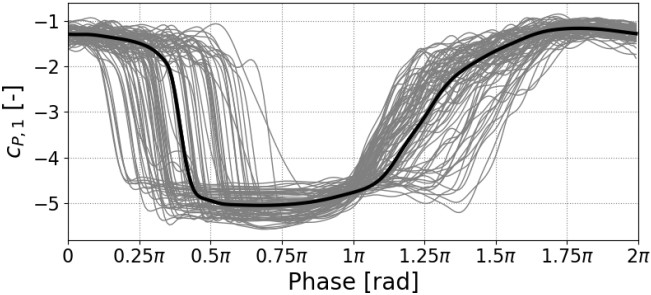

**Figure 3.** soft-DTW centroid for clustered time series with strong phase jitter. (Example data from pressure sensor reading from towing tank)

were taken to scale the process; firstly the data was down-sampled thus reducing "N" and secondly the code was scaled using DASK (Dask Development Team, 2016). DASK is a python library designed to parallelize standard python functions onto cluster architecture. The second step may at first appearance seem extreme, in practice the power required was more than a standard desktop but one or two compute nodes were more than sufficient. For the examples computed in this paper, 1-2 workers (nodes with 8 cores each) would process a single experiment within a few minutes. A combination of parallelization and downsampling was used in this study. [4]

Reducing the number of samples gives a significant speed boost as the complexity of the distance measurement is based on the number of time steps. While reducing the sample size, the spectral resolution is reduced about the same factor. The frequency of the expected phenomena limits the amount of downsampling. In order to improve the cluster results the data is, in addition to downsampling, filtered. Dynamic Time Warping is noise sensitive as the algorithm shifts and bends the time series in order to match similar values. Fortunately, tuning these steps is not difficult as a visual inspection of the resulting data will indicate whether the algorithm is making sensible groups or not. This topic is explored in greater detail in the related work from Steenbuck (2019).

Clustering is a method of dimensionality reduction based on the principle that the dataset can be efficiently described by a set of subgroups. These subgroups are formed on the assumption that the description of the cluster is a useful enough generalization for each member of the cluster. This means that the groups are formed on the basis of similarity. Clustering is an unsupervised method in the sense that there is no correct answer defined ahead of time. Usually unsupervised methods will reveal underlying data structures. This is not to say that we can just passively use these algorithms and useful results will ensue. Each clustering algorithm will perform well for some datasets and will deliver nonsense for others, care is required. To ensure good results, users will usually have to tune hyper-parameters for the dataset, the simplest of these parameters is the number of clusters. A first estimation about a reasonable number can be made from inspecting the dendrogram or the MDS plot as described below. Another approach is to calculate the mean silhouette score of all elements for a range of cluster numbers (Figure 6). The silhouette score is calculated by comparing the distance from a data element to its own cluster centre to the distance to the

---

[4]Combining the soft-DTW algorithm and the DASK module did require some programming effort, but as both tools were well developed, the effort was smaller than perhaps first appears. In particular, tools like DASK allow people with very modest programming skill to run cluster scale code.

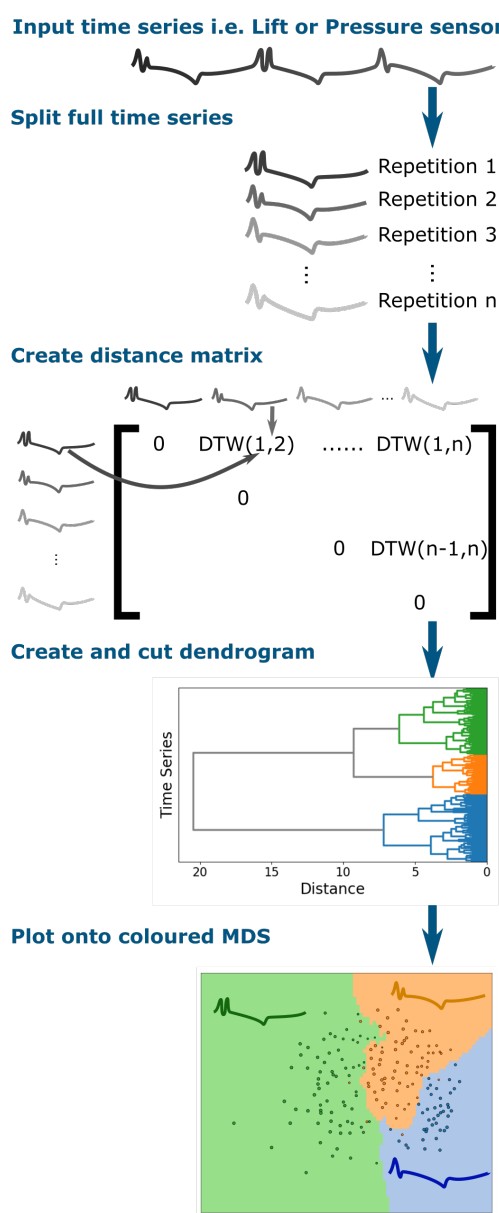

**Figure 4.** Time series clustering algorithm

centre of the closest neighboring cluster (Raschka (2015)). By calculating the mean silhouette score for a number of different clusters, we can see that once we get to 4 clusters, we only marginally change the quality of the clusters (shown in Figure 6). This means that breaking the dataset up into further smaller pieces is not going to improve our analysis.

For this application hierarchical clustering turned out to produce groups that were physically meaningful and shared features. Hierarchical clustering creates links between data points (in our case a single cycle of a dynamic stall test) to form a

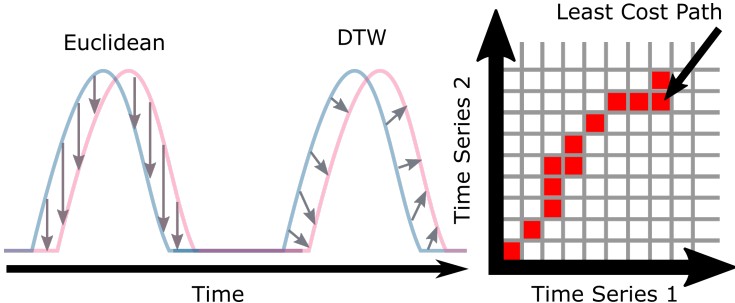

**Figure 5.** Euclidean distance vs DTW distance between two time series

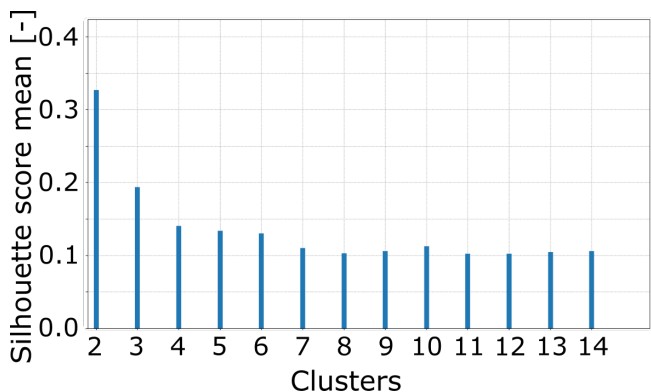

**Figure 6.** Mean of silhouette scores per cluster number.

dendrogram as seen in Figure 4. This process essentially takes the distances that we previously calculated and starts collecting similar data together recursively which is what is shown in the dendrogram. The dendrogram is then cut at a height which results in a given number of clusters. As longer branches indicate bigger differences, the height of cutting should be chosen so, that the longest branches are cut. The clustering was implemented using scipys (Jones et al., 2019) hierarchical clustering algorithm (scipy.cluster.hierarchy) with the ward method as a measure for distances between newly formed clusters. Hierarchical clustering was chosen after exploratory analysis showed that other basic algorithms such as KMeans tended to perform poorly for this data. Fortunately, well developed machine learning libraries such as Sci-Kit Learn, make it very simple to trial different algorithms.

Another way of presenting the data is to use Multidimensional Scaling (MDS) (O'Connell et al., 1999). MDS essentially takes a cloud of data points with high dimensionality and squashes the points onto a low dimension plane while attempting to maintain the distance between the points. In our case, each time step of a single series represents a dimension or feature which results in dimensionality that is incredibly difficult to interpret. Now take each series as a single data point and then squash it onto a 2D plane, and the data reveals an underlying structure. We can then color each point and use a k nearest neighbors classifier to color the background as seen in Figure 4. The resulting point cloud (hopefully) inherits distinct clusters.

The number of clusters encountered here, gives a good first estimation about a reasonable cluster number for further analysis. So instead of creating a chaos of overlapping time series, the data appears as a low dimensional representation image with each color representing time series with similar behaviour. In some circumstances, the coordinates of the image will even have a clear physical meaning i.e. dimension 1 could correlate with Reynolds number. A broad overview of the algorithm used in this paper can be found in Figure 4.

An example of the cluster analysis is depicted in Figure 4. In this figure, we summarize all of the time series of a single cluster by displaying only its centroid. We can see that each of the centroids represents a slightly different behaviours particularly during the secondary vortex shedding. Each cluster has a small uncertainty band shown by the standard deviation. As the dataset can be represented by three centroids instead of trying to compress the entire data into a single average, the representation is concise but still provides a more accurate view of the process.

## 3.2  Convolutional Neural Networks

In the previous section, we looked at how we can cluster together similar experimental samples. This section aims to see if we can extract some interesting features from our data using machine learning. For this example, we will attempt to use computer vision (machine learning applied to pictures), to extract information about the dynamic stall vortex.

Convolutional layers are the special trick that have turned neural networks into a wildly effective computer vision tool (Krizhevsky et al., 2012). Convolutional layers allow pictures to maintain their structure as a grid of pixels. Convolution operations are applied over the picture as a kind of moving window shape filter. The shape filters are learned and often end up resembling recognizable patterns. In the first layer of the network, the filters will be detecting edges, slow gradients and color changes (Zeiler and Fergus, 2013). As we proceed deeper into the neural network, the filters begin to look like natural features such as: a birds eye, a bicycle wheel or a door-frame. Each of these filters is created during the training process where large datasets are fed through the network and the error is propagated backwards through the network to allow for incremental improvement. It is helpful to note that as pictures are just made up of a grid of pixels, a 2D matrix structure (for single channel), in a great deal of cases, data can be represented in this form. This means that computer vision tools can be used on data that can be structured like a picture. Convolutional neural networks are most effective when features are local.

We have discussed here neural networks with a high number of layers. This is referred to as deep learning. Deep learning is a field that has seen rapid innovation due to the abundance of graphical processor units (GPU) and more recently tensor processing units (TPU). Platforms such as PyTorch or TensorFlow provide high level front ends in Python. These front ends abstract away much of the complexity meaning that users avoid much of the low level matrix algebra and optimization. Furthermore, it is common practice to publish well performing neural network architectures that are already pre-trained (transfer learning). Many of the once difficult decisions, such as choosing a learning rate, have now been made simpler with tools such as learning rate finder (Howard and Others, 2019). Cheap computational power, easy high level coding and the advent of transfer learning means that these incredibly powerful tools are now available for aerodynamic applications like detecting boundary layer transition from microphone data (see Figure 7). These innovations mean that non-machine learning specialists can use deep learning with a low barrier to entry.

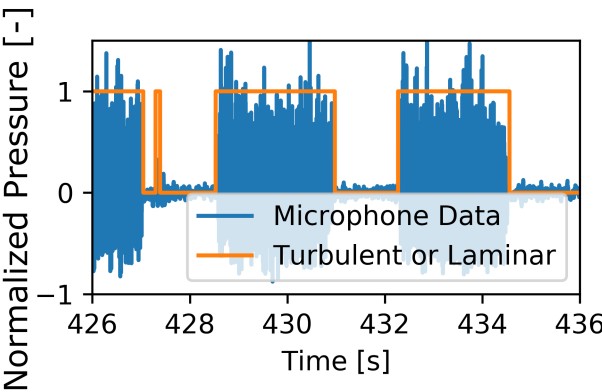

**Figure 7.** Identification of a boundary layer state using a recurrent neural network (data from Bak et al. (2010)) (see code example https://github.com/MatthewLennie/Aerodynamics)

In this paper, we will provide an example of turning aerodynamic data into a picture and then using a convolutional neural network to extract useful information. Dynamic stall vortices have a strong low pressure core which causes a lift overshoot and moment dump. When dynamic stall vortex data is averaged over 50+ cycles, it tends to show dynamic stall vorticity as far more clean and coherent than is the case for a single cycle. The strength of each vortex, its convection speed and onset of convection vary between cycles. This leaves the question: how much do dynamic stall vortices convect differently? Do boundary conditions like the reduced frequency affect the variability?

The dynamic stall vortex feature of a pressure vs time plot is easily distinguished by the human eye, however, pulling this feature from the data is rather difficult. The authors attempted the task with a number of more simple approaches such as simply finding the peak at each chord-wise position, a Hough transform, or even Bayesian linear regression with the pressure plot interpreted as a probability distribution. They all worked for a few cases but failed to generalize and in the end did not perform well enough to be usable. Each vortex is different and therefore manually creating a rule to automatically pull the dynamic stall vortex feature from the data wasn't trivial. However, this is a standard computer vision task very similar to a driver-less car identifying a cyclist in a picture. Fortunately, heavy development in the computer vision field has resulted in some incredibly powerful pre-trained models such as the RESNET family of models (He and Sun, 2016) [5]. The model is a convolution neural network that has been pre-trained on a massive dataset of real world images. This means that the convolutional layers of the network already have a set of shape filters that are broadly applicable to all natural pictures. This means that with a relatively small amount of training data and computational effort, we are able to simply re-mould the convolutional layers to identify dynamic stall vortices's and give the convection speed and phase.

Pre-trained neural networks can be built and re-trained using any of the typical frameworks such as PyTorch, Keras or TensorFlow. In this case, we used a RESNET50 model within the FASTAI architecture which is a high level interface built

---

[5]Note that while we were able to get acceptable results from the RESNET models, a higher level of accuracy may be obtained by network architectures that were built specifically for this kind of localization task.

on top of PyTorch (Howard and Others, 2019). The FASTAI architecture implements several current best practices as defaults such as; cyclical learning rates, drop-out, training data augmentation and data normalization. We can think of pre-trained neural network as a template, most of the training has already been done, we only need to retrain the network to react correctly to our dataset. This approach is cheap in terms of data volume and computational power.

The final layer of the neural network was replaced with two outputs to represent a linear fit of the vortex convection (slope, offset). For this analysis, acceleration of the vortex was ignored, though the code could be easily extended. The pressure data was represented as a picture where the horizontal dimension represents phase, and the vertical dimension represents the suction side of the airfoil with the bottom of the picture being the leading edge (an example of an already processed picture is in Figure 8, the training data does not have the blue line identifying the vortex but is otherwise the same). Training data was created by manually clicking (and storing) the positions of the vortex on 733 images (an attempt with only 300 pictures tended to over-fit on RESNET50 or have high bias on smaller models). The manual clicking does introduce some measurement error, but a few practice runs showed that the error was much smaller than the effect of the physical phenomena. The images were selected from a wide range of cases with randomized test training splitting within each case to ensure good generalization of the fitted model. However, data was limited to examples with a strong wake mode shedding meaning that the vorticity is easily visible on the pressure footprint. The training was done in two stages, first with the internal layers of the RESNET model frozen. This means we train only the very last layers that output the slope and onset. Once the training error reached stopped improving, the internal layers were unfrozen to mold the internal layers for a small number of epochs(training repetitions.

Initially 80% of the data was taken as the training set and the training was completed with 20 epochs with the convolutional layers frozen so that the newly added layers could quickly converge. The training was stopped at 20 epochs once the validation error begun to increase. The convolutional layers were then unfrozen the training was continued for a further 20 epochs. During training no geometrical augmentations on the training were undertaken but the brightness of the images was augmented [6]. The error statistics were still unsatisfactory and additional training did not improve the performance further. However, the current settings of the hyper-parameters settings and training procedure had seemed to extract the best model given the available data. The training procedure was repeated exactly the same a second time, with the same hyper-parameters and the same number of epochs, however this time the dataset wasn't split into test and training sets thus neglecting validation error (a practice described by Goodfellow et al. (2016)). This may be perceived as a opening up the risk of over fitting, however the training procedure and hyper-parameters were already tested and the neural network didn't over-fit. Furthermore, usually additional data will usually help reduce over-fitting. We therefore have confidence that with this procedure the validation error will not increase and that the training error is representative across the dataset. Thankfully the additional data did reduce the training error enough to make the model usable (see an example results in Figure 8). The residuals of both the slope and constant were distributed roughly as Gaussians with Standard Deviations of 0.15. In total, the training took in the order of 30mins of computational time on a GPU. Readers are encouraged to view the source code at "https://github.com/MatthewLennie/VortexCNN". The repository contains training sets and final data used to produce the following analysis.

---

[6]Geometric augmentations would have been the next method to improve the model if the process hadn't worked well enough

The resulting model incurs a small measurement error so the resulting distributions have be adjusted. Fortunately, the measurement error could be quantified. Both the error and the resulting vortex convection values can be approximated as Gaussian. The real distribution is sought by guessing a distribution, running a Gaussian convolution filter over the distribution and then measuring the difference between the resultant distribution and the data. Essentially, we knew the measurement error distribution roughly, we knew the output distribution, we can on a statistical basis work backwards. This error term is fed into a optimizer thus giving an estimation of the real data distribution without the error incurred by the neural network inference. In practice, this reduces the standard deviations of both the slope and intercept by roughly 30%. We should note that we can not "repair" the measurement data and locate the true convection speed of each measurement, but on a statistical basis, we can get closer to a true estimation. It is also worth mentioning that this neural net will find the speed that the vortex footprint travels across the airfoil, the vortex will usually have an additional component normal to the airfoil.

The procedure described above represents a first iteration of such an approach, a feasibility demonstration. With some more effort, a better neural network architecture could be chosen and the clicking procedure could be replaced with comparison to flow visualization. With these improvements, we could potentially avoid the final step where we attempt to repair the distributions. We would prefer to remove this final step which forces us to assume that the distribution is Gaussian. Nonetheless, the current model is workable enough for our purposes.

## 4 Examples

So far we have explored the idea that stall is variable as well as a few machine learning methodologies that could help interpret the data. We will now provide demonstrations of both of the algorithms. While the specific results are interesting and we will briefly discuss the physical effects observed by the algorithms, the aim of this section is to provide illustrative examples of the approaches in use. The description of the physical effects are provided merely to motivate that the methods appear to be finding sensible phenomena.

### 4.1 Extracting vortex convection with a convolutional neural network

In this section, we provide a demonstration of the neural network extracting dynamic stall vortices's from the surface pressure of the airfoil - time series data. The time series data comes from the wind tunnel dataset which sees an airfoil in a wind tunnel. The airfoil pitches sinusoidally, the free-stream velocity can be changed and the leading edge blowing is installed. A number of test configurations with dynamic stall were chosen and pushed through the neural network. The first case is relatively complicated, as it features, an oscillating inflow velocity (sinusoidal with a variation of 50% of the mean inflow), pitching into the dynamic stall range (up to $25°$) with leading edge blowing active. Four example tests were compared with different phase differences between the angle of attack motion and the inflow velocity. The pitch and blowing phases for each case are shown in Figure 9. Medina et al. (2018) made a very similar analysis and found that decelerating flow tended to destabilize the boundary layer and encourage earlier separation. With the convection speed and onset data retrieved by the neural network it is possible to show that this is true in the specific detail of the dynamic stall vortex. Figure 10 shows that for cases where the inflow speed

is in phase with the angle of attack, the shedding occurs later. However, when it does finally occur, the vortex will shed at a higher velocity (see Figure 11). Interestingly the results seem to indicate a much higher variability in the cases where the flow is decelerating during the vortex convection. Figure 14 also shows the relationship between the onset of the vortex shedding and the convection speed, there is a weak correlation ( 0.3 Pearson metric) but not strong enough with the existing data to make conclusions about the relationship between the two. This first example shows us that we can use a machine learning tool to better understand how our boundary conditions such as inflow velocity affect the physical process. We were able to take a large set of test repetitions and summarize them in a compact yet descriptive manner without having to resort to averaging.

A second example shows the effect of varying only the Reynolds number with constant inflow velocity (see Figure 13 and Figure 12). We can see that the mean vortex convection velocity scales with Reynolds number as we should expect. The vortex convection onset has a constant variance across both examples (see Figure 13). However, interestingly the variance of the convection velocity grows with Reynolds number (see Figure 12). This example shows us to be very careful about how we think about variability and how it applies to each part of the physical process. While these results and the first example's results are interesting and can be expanded upon, the important lesson is that a small data, low computational cost machine learning method was able to help extract a richer set of information from the dataset.

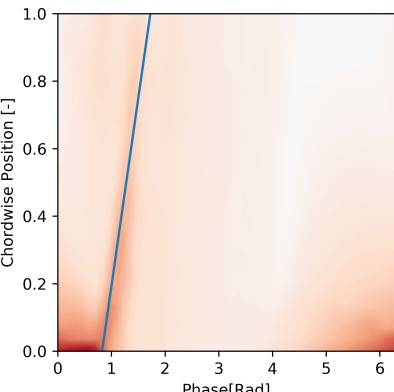

**Figure 8.** Example CNN Output, color intensity refers to suction pressure, blue line is regressed fit. Pressure is standardized therefore the colors represent Z-scores, no units or color bar are provided for this reason

-

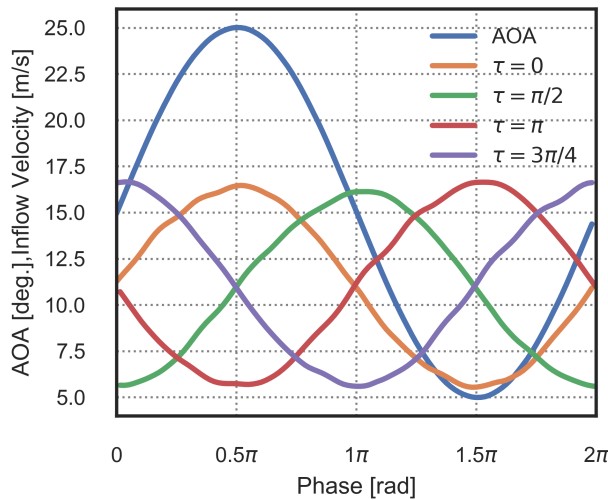

**Figure 9.** Inflow and angle of attack for Figure 10 and Figure 11

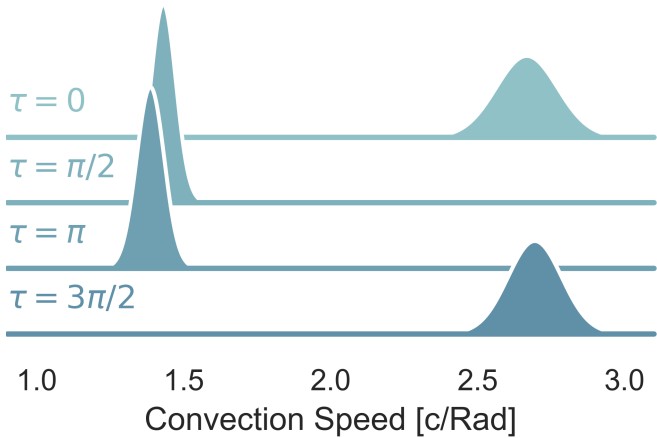

**Figure 10.** Probability distributions of the convection speed of dynamic stall with airfoil blowing different phases of harmonic inflow $(\tau)$. $\frac{U_{amp}}{U} = 0.5$, $k = 0.08$, $Re = 2.5 \cdot 10^5$ and $\alpha_0 = 15°, \alpha_{amp} = 10°$.

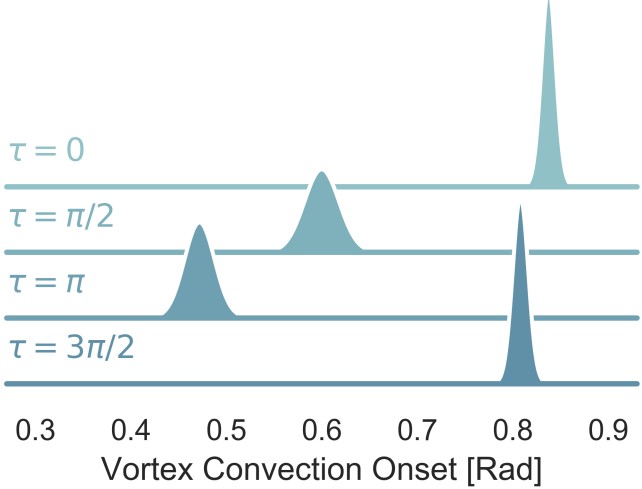

**Figure 11.** Probability distributions of the onset of dynamic stall with airfoil blowing and different phases of harmonic inflow ($\tau$). $\frac{U_{amp}}{U} = 0.5$, $k = 0.08$, $Re = 2.5 \cdot 10^5$ and $\alpha_0 = 15°, \alpha_{amp} = 10°$.

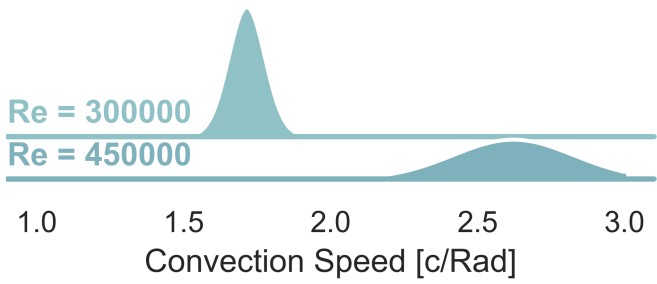

**Figure 12.** Probability distributions of the convection speed of dynamic stall with different Reynolds Number, $k = 0.09$, and $\alpha_0 = 18°, \alpha_1 = 7°$.

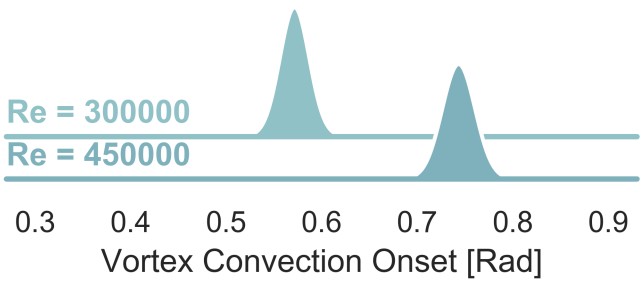

**Figure 13.** Probability distributions of the onset of dynamic stall with different Reynolds Number, $k = 0.09$, and $\alpha_0 = 18°, \alpha_1 = 7°$.

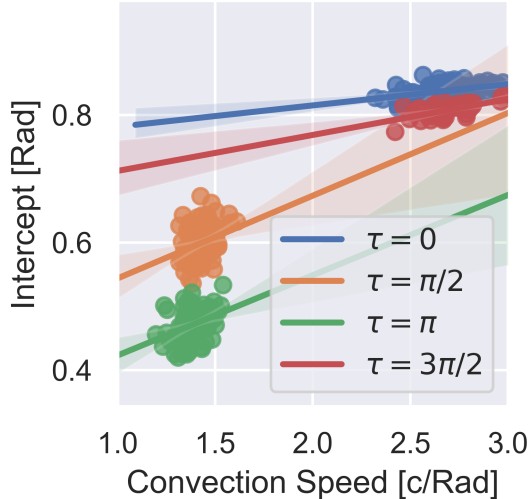

**Figure 14.** Relationship between the onset of shedding and convection speed for a range of blowing cases

## 4.2 Dynamic Stall Clustering

In the following section, we have chosen a few examples purely for the purpose of demonstrating the clustering approach and their usefulness in analyzing dynamic stall. In particular, we would like to see if there are distinct behaviors possibly stemming from stall cells or other complex phenomena. By using clustering, we hope to split our dataset into clusters of different airfoil behaviors as far as they exist. This provides us with a way of inspecting the data without having to laboriously compare each time series or to simple inspect averaged data that will hide these effects.

At high angles of attack ($\alpha_0 = 21.25°$ and $\alpha_{amp} = 8.25°$), we can observe the different kinds of stall behaviors that can occur. Figure 15 and Figure 16 show contrasting behaviors for the same angles of attack. In Figure 15, a quasi-periodic shedding appears. Without flow visualization it is hard to determine the shedding type, but the pressure footprint show the vortex as weak and smeared. This kind of footprint would indicate that the vorticity is not close to the surface of the airfoil or is large and not very coherent. This probably indicates that we are seeing a shear layer instability rather than very clear

wake mode examples seen in the previous section. The clusters seem to indicate that the shedding behavior is not reliable with cluster 3 (green) and cluster 4 (red) showing amplitudes of oscillation dissipating rapidly. Whereby, the other two clusters show a more sustained shedding pattern.

Now let us consider a second case with a different Reynolds number and reduced frequency but with the same angle of attack range (Figure 15). The airfoil moves into stall, releases one (cluster 2 - orange) or two coherent vortices's (cluster 1 -

440 blue cluster) and then resolves into weaker small scale shedding. That is, we are seeing two different shedding patterns for the same boundary conditions.

In Figure 17 and Figure 18 we can observe the effect of changing the reduced frequency while holding the Reynolds number and angle of attack constant. The first most obvious difference is that the period between the primary and secondary vorticity remain constant. The data does otherwise follow the general wisdom that the lift overshoot will increase with reduced

frequency, but it doesn't happen uniformly. Furthermore, the lower reduced frequency seems to create a much wider variance in the primary stall vortex compared to the higher reduced frequency where both clusters display a strong primary vortex with only a barely visible change in primary stall. Using the clustering method we are also able to reveal that, in both cases, one cluster has a strong secondary vorticity and the other has a nearly non-existent secondary vorticity (read carefully, colors do not match). Interestingly the higher reduced frequency in Figure 18 seems to suppress the secondary vortex as Figure 17 shows

strong secondary vorticity 55.8% of the cycles and a somewhat weaker secondary vortex for the other cycles.

We have observed with these four example cases that differences in reduced frequency and Reynolds number will resolve into a quite different type of vortex shedding. Furthermore, even within the same case we can see a strong variation in the strength of the shedding mechanism. The instability mechanism driving this shedding is very sensitive to the small variations in input conditions. The shedding mechanisms shown in these four examples are just one of the variety of shedding behaviors.

A quick visual inspection of the time series data would be unlikely to uproot the variable shedding behaviors seen in these two examples. However, the cluster centroids or even simply the MDS plots (i.e. Figure 20) make the differences clear and easy to interpret. In any of the example cases, phase averaged results would have been a poor representation of the dataset

because we would not have any way of seeing the variable nature of the results. Better information leads to better decisions, for example, when we calibrate simulation tools, particularly empirical unsteady aerodynamic models, we should be aware of where our models will perform poorly because the underlying flow physics is highly stochastic even showing distinct behaviors. Our model design choices can be more well informed, i.e. choosing to fit a model on the most commonly occurring cluster only or even trying to recreate the variability. We should also be aware that standard measurements of a models performance such as mean squared error are only valid for homoscadastic regimes, that is, we expect the same amount of variance throughout the whole range of the models validity. If we violate this condition, the models will tend to be a poor representation of reality. This is true for fitting machine learning models and also the semi-empirical models commonly used in unsteady aerodynamics. Finally, one can easily find examples of experimental field data where clustering would be a powerful data analysis tool, e.g. the double stall measurements from Bak et al. (1998).

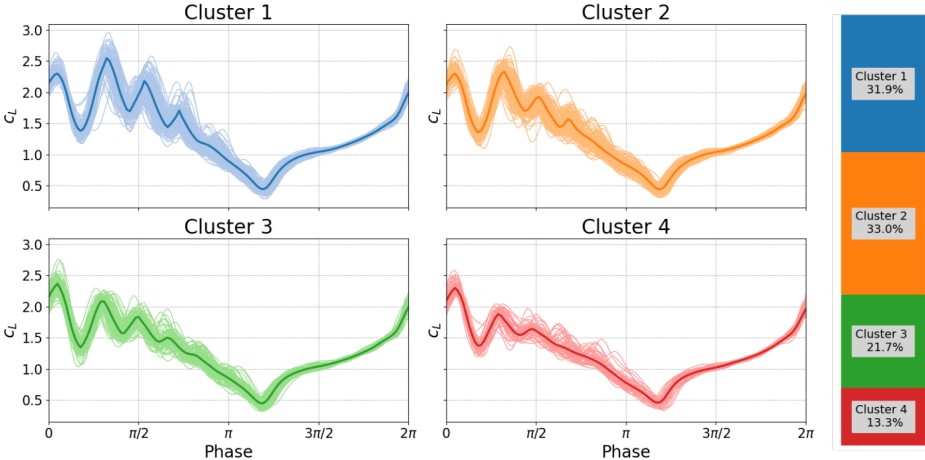

**Figure 15.** Deep stall investigations: cluster analysis for boundary conditions: $k = 0.0992$, $Re = 3.3 \cdot 10^5$ and $\alpha_0 = 21.25°, \alpha_{amp} = 8.25°$.

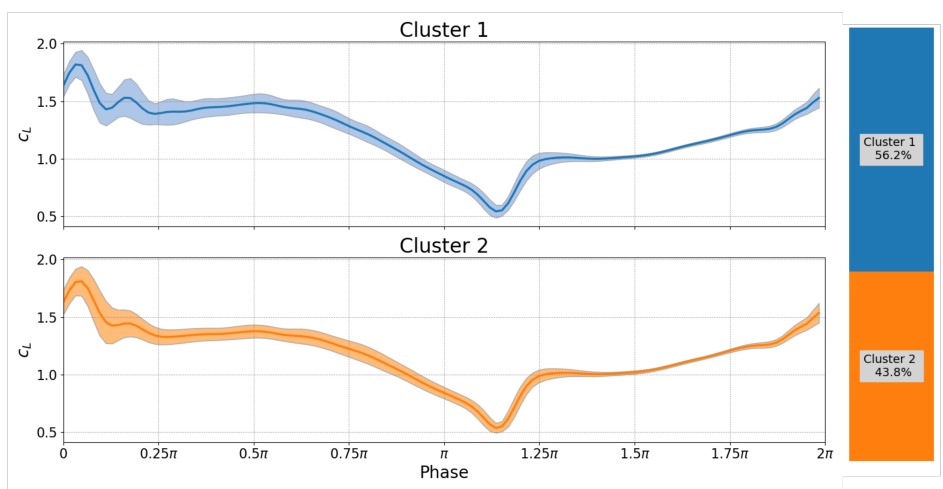

**Figure 16.** Deep stall investigations: cluster analysis for boundary conditions: $k = 0.0574$, $Re = 5.7 \cdot 10^5$ and $\alpha_0 = 21.25°$, $\alpha_{amp} = 8.25°$.

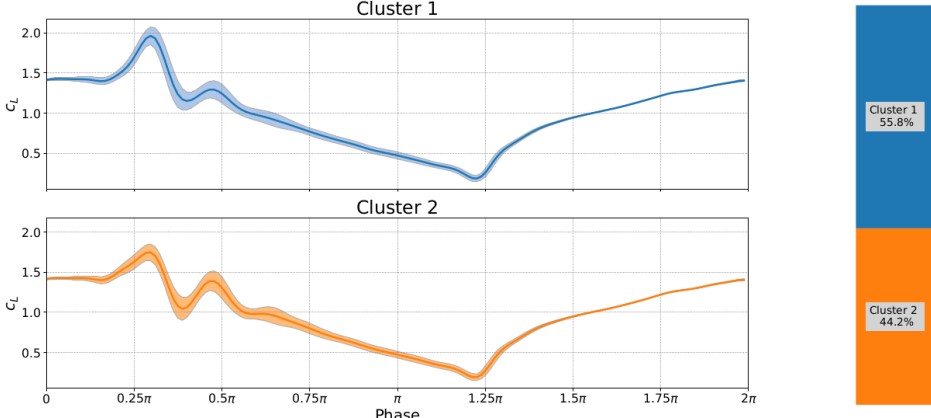

**Figure 17.** Deep stall investigations: cluster analysis for boundary conditions: $k = 0.0992$, $Re = 3 \cdot 10^5$ and $\alpha_0 = 18°$, $\alpha_{amp} = 7°$.

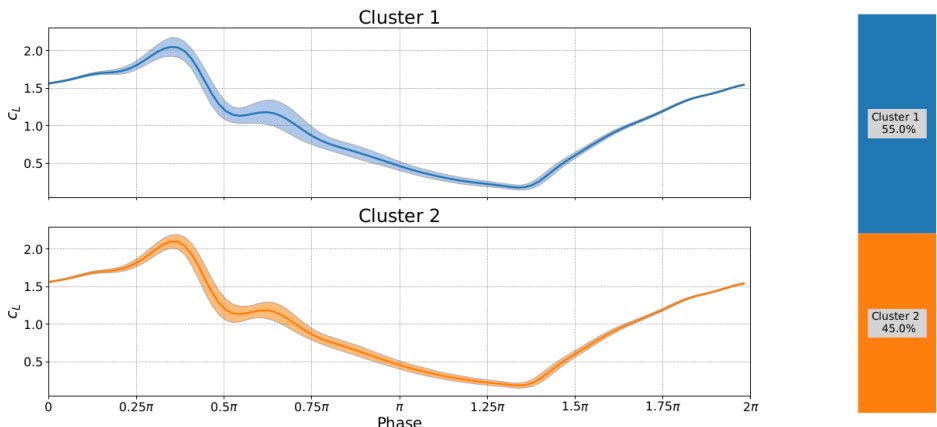

**Figure 18.** Deep stall investigations: cluster analysis for boundary conditions: $k = 0.1346$, $Re = 3 \cdot 10^5$ and $\alpha_0 = 18°, \alpha_{amp} = 7°$.

## 5 Convergence and outliers

The clustering and MDS can also be used together to qualify outliers that may corrupt the quality of the dataset. For instance in wind tunnels, the first cycles of a test will often be different to later cycles due to the wake effects and dynamics of the tunnel. Similar start up effects can also be seen in the towing tank. However, more broadly speaking, test data are often plagued with test data poisoned by some sort of external influence. Figure 19 is an example of a single leading edge pressure sensor from the towing tank where obvious outliers are present. The pressure values in the main cluster (blue) show detached flow over the entire cycle. However, a small number of cycles in the green and orange clusters actually reattach. The MDS representation alone (Figure 20) indicates that it is worth inspecting the data further. Such an obvious representation could speed up the task of possibly pruning the dataset where outliers are created by known effects such as startup or a measurement failure.

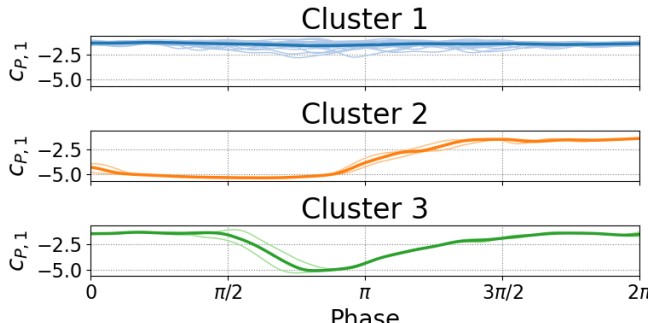

**Figure 19.** Clustered time series from towing tank surge experiment. Boundary conditions: $U_\infty(\phi) = 2.5\,\mathrm{m\,s}^{-1} + 0.7\,\mathrm{m\,s}^{-1}\sin(\phi)$, $\overline{Re} = 1.25 \cdot 10^6$, f=0.21 Hz, $\alpha = 10°$

It would also be possible to remove outliers automatically based on the cluster data. In practice, this level of automation is not necessary on most experimental setups and the visual inspection provided by MDS and clustering was enough to find outliers quickly and efficiently. On a practical level, it is possible to put the MDS plots into a folder and view the image thumbnails an efficient quality assurance step.

While in this paper we have broadly recommended making cluster based centroids rather than a mean of the whole dataset, the reality is that the latter is still common practice. McAlister et al. (1978) made the recommendation of taking at least 50 cycles of data to ensure convergence of cases with dynamic stall. The methods used in that paper were limited by available computational power.

Bootstrapping is a method of uncertainty estimation which uses re-sampling. The concept is quite simple; stick the data in a bucket, re-sample with replacement until you reach the size of the dataset, then find your mean, variance and other statistics required. This process is then repeated until a probability distribution of the values is found; very similar to the concept of confidence intervals. This provides us a quantitative statement such as "the existing data indicates 90% of the time that the mean lies between 0 and 1". Bootstrapping has some nice mathematical properties mostly propagating from central limit theory. A good treatment of the subject is given by Chernick (2008).

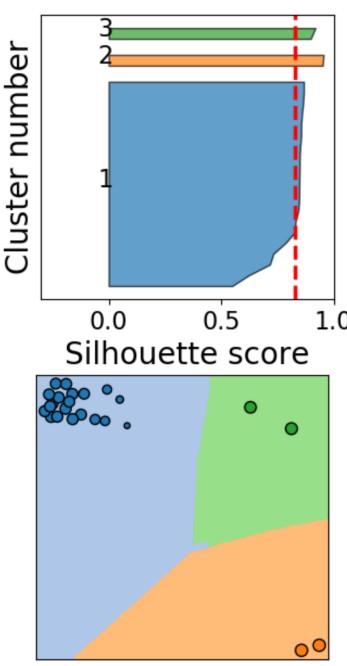

**Figure 20.** Above: Silhouette Samples per cluster. Below: MDS representation. Data from towing tank surge experiment

In our case, we would like to see how the uncertainty of our population estimates decrease as we collect more data. To do this, we repeat the bootstrapping process pretending at each step that we only have a given number of cycles. This results in a graph comparing uncertainty to number of cycles available (see Figure 21 and Figure 22). One will note that the variance and inter-quartile ranges converge slower than the mean and median. This is due to the simple fact that the central moments of the distribution will collect more data more quickly, and will therefore converge with less data. In practice this means, how much data you need will depend on whether you need the central moments or the extreme events.

Lennie et al. (2017) demonstrated that when considering stall, it is probably best to avoid using mean and variance due to the non-Gaussian spread of the data. Median and inter-quartile range will serve better in cases of stall. All of the population estimates are presented here, as percentile based estimates such as median and inter-quartile are still rarely used in literature. Representing the variability with a non-parametric distribution (Kernel Density Estimate) gives the best representation and can be achieved with violin plots (see examples in Lennie et al. (2017)). The error itself is based on the temporal mean of the respective estimate throughout the time series. A similar convergence approach was used in (Lennie et al., 2018a).

A number of test cases were chosen with varying degrees of separation. In deep stall cases, as seen in Figure 22, the error of the standard deviation drops below 2% after 60 repetitions. The light stall case in Figure 21 shows quick convergence at low values. Already after 20 repetitions all errors are below 1%. In cases with unsteady inflow, the normalization of aerodynamic coefficients with the inflow speed can amplify experimental noise and therefore converge slower than expected. It may be possible to converge the inflow speed and lift values separately then apply normalization to speed up convergence. Of course

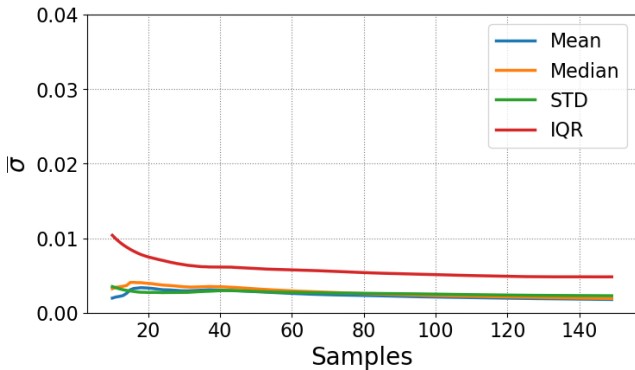

**Figure 21.** Convergence of the population estimates for a light stall case as the number of tests increase

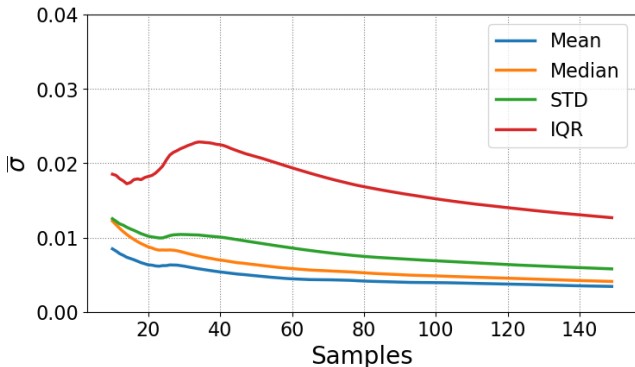

**Figure 22.** Convergence of the population estimates for a deep stall case as the number of tests increase

different levels of confidence would require more or less repetitions, however, for general purpose the following principles can be made:

1. For deep stall use <60-100 Cycles.

2. For light stall use <20 Cycles.

3. Be careful in cases with unsteady inflow, even attached flow can take up to 40 Cycles to converge.

These principles should be read in context of the limited example given here. In most of the examined cases, the variability and thus the rate of convergence was reduced with higher Reynolds numbers. The higher the angle of attack, the more pronounced
the effect. The convergence may be influenced further by the reduced frequency and the addition of flow control elements. It is always best practice to conduct the bootstrapping for each new test configuration.

While a main recommendation from this paper is to use clustering to represent data, simple averaging will remain a popular analysis tool. However, we advocate using bootstrapping to at least help quantify the uncertainty of the averaging and clustering

to find outliers. Even if the final analysis will be conducted on averaged data, the steps outlined in this section will still help isolate problems with the dataset. [7].

## 6 Potential new dynamic stall modeling approaches

The marriage of data science and aerodynamics presented in this study has so far been an exercise of data visualization. However, machine learning tools can also be useful for other tasks such as robust dynamic prediction (Brunton and Noack, 2015). The natural extension of this study would be to create a new generation of unsteady aerodynamics models using machine learning techniques. A challenge to the current stable of unsteady aerodynamic models is modeling vortex induced vibrations. Wind turbines can be exposed to very high angles of attack particularly during construction and shutdown and furthermore the blades are relatively flexible giving rise to vortex induced vibration problems (Lennie et al., 2018b). There are complex relationships between the operating regime and the wake structure (Lennie et al., 2018b). The empirical models[8], such as the Beddoes-Leishman model, will be very difficult to extend to handle vortex induced vibrations given the fact that shedding behavior varies strongly with many of the input conditions and therefore will be hard to encode into a readable set of equations. The authors of the Beddoes-Leishman model even hint that the model they developed were difficult to extend without amplifying noise (Leishman, 1988). Essentially it becomes too difficult for a human creator to write down a complex enough model that is well behaved over all operating regimes. This is a recognized problem in machine learning, that models with enough capacity to learn a complex system tend to memorize the training data and perform poorly on new examples. This is called overfitting.

Machine learning provides another path to improving aerodynamic models, as it provides the tools and techniques to fit high capacity models while simultaneously handling the problem of over-fitting. Such an approach would perform much the same roll as the current models but would be machine learned. It is important to note here that neural networks are not a look up table. In the same way that our convolutional neural network learned more complex features as we moved deeper into the network, a neural network would begin to learn abstractions that are useful in the context of unsteady aerodynamics i.e. relationships between angle of attack and lift.

Of course, the network has to be trained to learn these abstractions. Using the concept of transfer learning it would be possible to train the model in stages. We outline a potential recipe for creating such a new model with the disclaimer that this is speculative and we fully expect some of the stages to require modification. Nonetheless, the recipe discusses the principle of training in stages, first with larger quantities of computationally cheap data and then again with smaller quantities of higher quality data. The machine learned model training process could be achieved with the following steps:

1. Generate a huge set of "cheap" training data using a standard unsteady aerodynamic model.

2. Train the machine learning model on this data until it performs as well as the standard model.

---

[7]An extended set of results can be found in the Master's Thesis of Steenbuck (2019)

[8]Holierhoek et al. (2013) have a good comparison of the models

3. Generate unsteady CFD and experimental training data for a single airfoil.

4. Use the smaller amount of higher fidelity data to further train the machine learning model.

5. For each airfoil, generate a small amount of CFD data

6. Recalibrate the machine learned model to each airfoil.

This approach has the advantage that the model can be constrained with a near endless supply of cheap data from the standard unsteady aerodynamics models. We would now have confidence that over nearly all operating conditions the model wouldn't diverge too far from reality. In this first stage, we have trained the network to learn a useful set of abstractions that apply to unsteady aerodynamics. The model can then be re-moulded just enough to represent the higher fidelity data from experiments and CFD without losing the constraints set in the previous step. This would produce a base model. For each new airfoil a new sub-model could be spawned off with a small amount of training data and computational effort. This means we have the robustness of the engineering model with an improved ability to match high quality data.

This concept does come with some challenges. The current low fidelity unsteady aerodynamics models are not designed to produce results at very high angles of attack. Furthermore, at very high angles of attack it is usually required to use 3D CFD to get high quality results. Finally, the shedding modes are affected by the flexibility of the structure, that is to say the full 3D structure. A possible approach is to use very rough approximations for the cheap training data (just based on the Strouhal number) followed by 2D CFD. While these two approaches are unlikely to be accurate, it will pre-train the model to reproduce the rough physics. This would then reduce the amount of 3D CFD with structural interaction that would be required to represent the very high angles of attack. This approach would treat the final version of the model as a blade dynamic stall model rather than an airfoil. These simulations would still require large amounts of computational power given current standards but will be the cheaper (if not cheap) approach. While the method described here doesn't provide a final approach, it hopefully demonstrates a useful machine learning principle of refining models in stages to make the best use of the data available.

## 7 Conclusions

This paper has attempted to bridge the gap between unsteady aerodynamics and the field of data science/machine learning. In particular we have attempted to provide some use cases of machine learning in unsteady aerodynamics. Stall is a complex phenomena which varies in both time and space and the data has shown strong variations between cycles of the experiments. The combination of clustering, dynamic time warping and multi-dimensional scaling allow us to effectively cluster cycles together making the data easy to interpret and reveal patterns that were previously difficult to inspect visually. Convolutional neural networks allow us to use computer vision on pressure data to find dynamic stall vortex convection. Using neural networks to extract complex features from data has an incredible potential with-in aerodynamics especially due to the advent of transfer learning.

Even the few examples analyzed in this study demonstrate that stall behavior is complex. The clustering results demonstrated that the shedding behavior varies across cycles especially in the secondary and tertiary vorticity. The neural network was able to extract the vortex convection feature from the pressure plots to show that the onset of dynamic stall and the convection speed vary with the inflow conditions as well as cycle to cycle. The approaches described in this paper are just examples of the potential approaches that can be used to provide detailed insights into unsteady aerodynamics data.

The results of this study already provide a number of recommendations about stall and data science.

1. Means aren't a sufficient description of stall. Data science and machine learning provide good ways of investigating cycle to cycle variations.

2. Multidimensional scaling and clustering with DTW as a distance metric is an effective way of examining data for different shedding modes or experimental outliers.

3. Dynamic stall behaviors vary significantly even within the same test conditions.

4. It is unlikely the traditional empirical models are the solution to modeling stall more accurately, machine learning may be the better option.

5. Dynamic stall vortices's will convect at different times and with different speeds. A neural network can retrieve this information from pressure data with a reasonable amount of training data and computational resources.

6. The bootstrapping method will help with determining the number of cycles needed to reach a given level of confidence.

7. The examples in this paper didn't require huge datasets (though can be used on larger datasets) or large computational resources, nor did it require significant amounts of specialized knowledge.

Finally, we hope that the demonstrations provided in this paper will communicate that there is a rich family of machine learning methods available for use in wind energy, unsteady aerodynamics and other adjacent fields.

## 8   Code Availability

A distillation of the codes used in this paper are available on https://github.com/MatthewLennie/ The data used for the convection plots is also in the repository. An example file is provided for the time series clustering with MDS plot.

## 9   Author Contribution

Matthew Lennie prepared the manuscript with the help of all co-authors. The computer vision model was constructed by Matthew Lennie. Johannes Steenbuck constructed the clustering code with assistance and supervision from Matthew Lennie. Bernd R. Noack provided code review and technical advice. Christian Oliver Paschereit provided assistance with the paper review.

## 10  Acknowledgments

The wind tunnel data was taken by Hanns-Mueller Vahl and David Greenblatt at the Technion - Israel Institute of Technology in conjunction with the Technische Universität Berlin. The towing tank data was measured by Marvin Jentzsch and Hajo Schmidt at the Technische Universität Berlin. The time series microphone data used in the recurrent neural network example was provided by Helge Madsen under the DanAero project. The DanAero projects were funded partly by the Danish Energy Authorities, (EFP2007. Journal nr.: 33033-0074 and EUDP 2009-II. Journal nr 64009-0258) and partly by eigenfunding from the project partners Vestas, Siemens, LM, Dong Energy and DTU. These datasets required extensive measurement effort and on going consulting to make the data useful, for this the authors acknowledge and thank the contributors. The authors would also like to thank Kenneth Granlund and George Pechlivanoglou for providing interesting feedback. The researchers would also like to acknowledge the support received from NVIDIA through their GPU Grant Program.

## 11  Competing Interests

The authors declare that they have no conflict of interest.

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
