# Peer review of "Cartographing dynamic stall with machine learning"

_Wind Energy Science, 2019_

## Referee Comment (RC1) · Anonymous Referee #1 · 4 Sep 2019

Dear authors,

I think the article is quite interesting and it seems that the machine learning techniques can indeed be used to analyse dynamic stall phenomena. In my opinion, the article should be shortened by removing some Figures and potentially also removing the towing tank experiments that are almost not used in the article except for showing an outlier (if I understand correctly). I find it very hard to judge how relevant the work presented in this article actually is. The authors suggest that the presented machine learning work flow can be used to analyse CFD results or experimental input. The results from this analysis could then be used to further train a model that is based on a massive amount of cheap data from empirical models. This might be very difficult due to the following reasons:

- The cheap models do not contain any unsteadyness for steady state inflow at constant angle of attack. This unsteadyness is exactly what would be needed to for example use a dynamic stall model for vortex induced vibration in deep stall.

- Since this unsteadyness is not included in the empirical models, all the necessary information needs to come from CFD or measurements.

- Vortex induced vibrations are a 3D phenomenon and depend on the blade shape and the aeroelastic mode shapes. So 2D computations or measurements will probably not be sufficient and the models would have to be trained on full blade CFD or measurements.

- This doesn't seem to be feasible due to the very high cost of training data.

- The dependency on inflow turbulence and stall delay/rotational augmentation increase the amount of necessary training simulations / measurements also for normal operation.

So I am not sure how realistic the training procedure outlined in the conclusions actually is.

More specific comments follow below.

Figures:

- Sometimes the Figures are placed far from where they are referenced.

- Figure 4 contains Figures 7 and 8 and thus Figures 7 and 8 should be removed from the article.

- Does there need to be an additional Figure 9 with an example of the clustering when there are plenty of figures with clusters later on in the article?

- Are Figures 10 and 11 necessary for the article?

- A color map for the pressures in Figure 12 is missing.

- I can't find a reference to Figure 13 in the text. Is it necessary?

- Plotting Figures 14 to 16 each on 'regular' line plots with one y-axis (instead of 4 y-axes) would probably be more clear.

- Some Figures, for example Figure 18, have very large font sizes. Please adjust.

- The two clusters in Figure 22 look virtually identical. This should at least be commented on in the text.

- You could consider plotting the clusters in a single plot so that the differences are easy to see, for example in Figure 21.

Other points:

- Please change the first sentence of the abstract: 'Airfoil stall is bad for wind turbines.'. Without airfoil stall limiting the lift coefficients, the extreme loads of the turbines would be much higher, so this sentence is simply not correct.

- It seems that the towing tank profile is only instrumented on one side, see Figure 2, and thus dynamic lift coefficients can't be computed from pressure integration of the surface. I can see that the towing tank experiment is used for the outlier detection in Figures 23 and 24. But maybe such an outlier detection could be performed on wind tunnel experiments instead and then the towing tank experiments could be removed from the article. At least for me it is difficult to see the value of these experiments in the context of dynamic stall.

- You write on page 3: 'In short, this phase difference can lead to single degree of freedom pitch flutter also known as stall flutter.' To my knowledge, the term 'stall flutter' is typically used to either describe instability due to vortex shedding (see for example 'a modern course in aeroelasticity') or so-called stall induced vibrations (vibrations with negative damping due to negative lift gradient (see for example Hansen 'Aeroelastic instability problems for wind turbines')). Both of these phenomena do not occur due to dynamic stall phase lags. For stall induced vibrations, dynamic stall can even reduce or reverse the effective lift gradient and thus stabilize an otherwise unstable operating point.

- Also on page 3: 'If the unstable nature of separated flows leads to the extent and phase of light stall to be variable between cycles of pitching, then it follows that the aeroelastic stability of the airfoil will also be variable between cycles.' Typically stability refers to the stability of an operating point. Thus if on average many cycles remove energy from vibrations the operating point is stable, if they add energy then it is unstable. Stability is not determined from cycle to cycle. Otherwise operation in turbulent inflow would switch between stable and unstable all the time.

- On page 8 you write '1-2 workers (nodes)'. How many cores per node?

- A large part of Section 4.1 is actually a description of the method, not a presentation and discussion of results. As such it should be part of section 3.

- On page 15 you write: 'The first case is relatively complicated, as it features, an oscillating inflow velocity, pitching into the dynamic stall range and leading edge blowing.' I think it would be a good idea to start with a much less complex case.

- On page 18 you write: 'Interestingly the higher reduced frequency in Figure 22 seems to suppress the secondary vortex with only 23.5% of the cycles having a secondary vortex where as Figure 22 shows strong secondary vorticity 51.8% of

the cycles.' It can't be Figure 22 you are referring to twice and I can't find the numbers 23.5% and 51.8% in the Figures.

---

## Referee Comment (RC2) · Tuhfe Gocmen (Referee) · 6 Sep 2019

Dear Matthew and co-authors,

Your work focuses on bringing the accelerated developments in data analytics to the investigation of the dynamic stall behavior. Hence, it is an extremely interesting and innovative study, also very-well written. Although the turbine aerodynamics is not my major field as you might know, the article was easy to follow and the reader is guided with interesting questions raised occasionally. Having said that, this particular review of the article will focus almost exclusively on the deep learning methodology presented and implemented, as well as the overall structure of the manuscript.

[Figure]

Section 2 - Experimental Data:

- The online database for the wind tunnel dataset is nicely referred in the text. However, the reader should be able to tell which channels/signals from the dataset and at which resolution are considered in the training and validation of the neural network, without having to download the .zip files. Additional comments on the clarity of input data and pre-processing will follow in the corresponding sections of the article.

Section 3 - Machine Learning Approaches:
- The implementation of DTW is super interesting, and the brief description is nicely supported with Figures 3, 5 and 6. Note that the font size for Figures 5 and 6 are too big, and they can probably be merged into one Figure only.
- Figure 4 is a nice flowchart describing the methodology for clustering. Several comments about that:
1) The font is again too big,
2) It is better placed later in the article in my opinion, when all the pieces come together (now we see it before we read anything about dendograms for example)
3) The very first item on this flowchart can be extended to show the exact input variables, including the look-up tables, if any. It is rather unclear what exactly is fed into the clustering algorithm. It can also be a small table to support the flowchart, if Figure ends up being too crowded...
- In terms of pre-processing the input data, you have indicated the use of down-sampling and filtering. It would be nice to see a discussion on the time scales those processes might be affecting in your dynamic problem.

Section 4 - Results:
- As indicated in the previous review, Section 4.1 is still the methodology and it should probably be a part of Section 3.

- No need to say, it is simply exciting to see transfer learning through CNN applied to turbine aerodynamics. It would be nice to see the final architecture of the network though, including the frozen layers, and the number of parameters to train in two newly added layers. You can probably replace Figure 10 (which is generic) with that (which is specific to your problem).
- You have my sympathies for manual clicking on 733 images... How to quantify the uncertainties in the input then? :)
- The arguments regarding over-fitting problem are surely valid, can also be supported by presenting of data / of parameters to fit ratio (somewhat more quantifiable).
- Would you elaborate on the post-processing of the model results a bit more (on Page 9, Lines 85-95), especially since there is a significant reduction in the variance? Would be nice to see 'before-after', where the original distributions of the model output is presented together with the 'corrected' distributions.

Section 6 - Conclusions:
- Maybe the 'possible deep learning applications' part should be rephrased, can be tricky to find unsteady low-cost models for many wind energy applications, as Reviewer 1 also mentioned. Why not further train the generated model of this study? You can also maybe discuss how/if your network can be expanded by introducing data of a 'different texture', e.g. images vs. numeric time series, different resolutions, etc.

General Comments
- Most of the issues regarding Figures are addressed by Reviewer 1. One additional might be the legend placement on Figure 18 (the blockage effect...)
- There are typos but very few, here are the ones I managed to catch:
1) Page 4, Line 95 'blowing..' –> 'blowing.'
2) Page 4, Line 98 '/data-resource .' –> '/data-resource.'
3) Page 10, Line 14 'too' –> 'two'
4) Page 11, Line 20 'the' –> 'then'

5) Page 12, Line 72 'Dominin et al.' –> 2019 missing?
6) Page 15, Line 55 (TODO) ??? :)
Don't just take my word on it though, have another round of read.

Thank you for your super interesting work, looking forward to the updates!
Many greetings
* * *

---

## Author Comment (AC1) · 4 Nov 2019

Author Comments.

When a WES paper is published, the journal editors completely retype set the paper. They rescale figures etc.. All of the figure sizes are intended for the final paper rather than the draft. For these reasons, the figures are all over the place etc.. The final visual presentation of the paper is determined by the journal, not me.

**Referee #1**

Thank you for the detailed comments on the paper. It may appear in the following section that I am hostile to your comments, this is not the case, I appreciate the feedback even if I have responded in an argumentative fashion in the following sections.

Please note that while the experiments mostly inspect dynamic stall, my literature review places dynamic stall in the wider context of unsteady aerodynamics.

**Comment regarding training method.**

I would like to start with the topic of a machine learning approach to unsteady aerodynamics. For my description to make sense it is important to understand that methods such as neural networks are trained in steps using stochastic gradient descent. In short, this means that we put a data point into the network, we then measure the difference between the correct output and the output of the current configuration of the model. We then slightly nudge the model in the direction of creating the correct output. A deep learning neural network architecture has enormous flexibility, while this is great because we can approximate any function, by definition most of these functions will be the wrong function. To constrain the neural network from all possible functions down to something that looks like an aerodynamic response of an airfoil, we need a good deal of data. As you rightly point out, it would be silly to attempt this with CFD.

Here is the important part of the argument. Because we can use stochastic gradient descent to train the model, we don't need to use the same input data over the whole training process. We can use low resolution data in the early training stages to bring the neural network into the correct neighbourhood. Out of the all of the possible functions that a neural network can create, the poor quality approximation of the aerodynamics which lacks turbulence etc. is still much closer to CFD or field data than all possible outputs. Is it a good model at this stage? No. But it is relatively well behaved, it could do a nice impersonation of the Beddoes Leishman model for instance.

What now? Once we have a model that contains something that looks roughly like unsteady aerodynamics, we can begin refining the model with a much smaller amount of higher quality data. To use a metaphor, the poor quality data got us into the neighbourhood like a bad quality street map, the better quality data is like a good quality street map of a very limited area, it helps us find the right street.

The ideas that I have presented here have a precedent. Jeremy Howard was able to win a neural network image identification challenge using much smaller amounts of computational time by first training the neural network on 16bit images and then only training on 256 right at the end.

The important question is of course: will this all work? My answer is, I don't know. 3D effects, different time scales all these things will make it a difficult model to train. However, I believe that the innovations of modern machine learning approaches have opened this new avenue of model building. My exploration of this topic was intended communicate the existence of this new avenue rather than prescribing an exact solution to unsteady aerodynamics.

**Comment Regarding Airfoil stall is bad.**

I understand your point but I disagree.

1. Stall reduces lift. We see vortex generators in the inner and outer regions of wind turbines to mitigate this effect for increased power production and noise reduction (for this I only have anecdotal evidence admittedly)
2. Stall causes phase lags which can result in instabilities. ( I will come back to this point)
3. Review papers such as Holierhoek, J. (2013). An Overview of Possible Aeroelastic Instabilities for WIND ENGINEERING, 44(0). Make a very long list of instabilities, a great deal of them being stall driven.
4. While smaller wind turbines were stall regulated, the design methodology of modern wind turbines uses pitch and variable speed to regulate wind turbines. In part due to the instabilities highlighted in point three. In short, modern wind turbines seem to be designed to avoid using stall as a regulation of lift which seems to be reflect that on balance, wind turbines stall is bad.

While there may be certain load cases where as you pointed out, stalling of the blades will reduce extreme loads, in the very next sentences I provide the context for the statement. While you do make an interesting counter point, I would prefer not to change the sentence. With respect, the boundary case you present does not present a strong enough argument to say: "simply not correct". It's not a critical premise in a syllogism, I believe that the balance of the argument is on my side, and I believe that it is a succinct way open opening the topic.  I would happily hear an expansion of your point, perhaps I have missed your point.

**Stall Flutter**

I will clarify this point with a few references to literature, because their explanations are far better than mine. Stall flutter doesn't imply dynamic stall. In fact, while you may get a large response from deep stall, light stall will tend to drive down the aerodynamic damping in the torsional direction. If you look in the figure below you can see the hysteresis loops have much larger negative damping than the deep stall case, you alluded to this effect in your comments. If you read the paragraph carefully I clearly state that I am talking about light stall. In page 303 of the reference, the distinction is made between stall flutter and dynamic stall, though it is my experience that stall flutter can be applied as a named to many many different phenomena. I believe my description was fairly careful.

[Figure]

*Figure 9* Unsteady forces and moments in three dynamic-stall regimes: $M_\infty = 0.3$, $\alpha = \alpha_0 + 10° \cos \omega t$, $k = 0.10$.

McCroskey, W. J. (1982). Unsteady Airfoils. *Annual Review of Fluid Mechanics*, *14*(1), 285–311. https://doi.org/10.1146/annurev.fl.14.010182.001441

Regarding phase shift, remembering that work or the energy absorbed or dissipated is defined as the boundary integral of these loops.

$$W \equiv \oint M_R \, d\alpha_R$$

This is just one way of looking at the equations. Another way is the decomposition of the lift into complex values in the same manners as Theodorsen. Carta and Niebanck sum it up neatly.

and after Eqs. (15) and (16) are substituted into Eq. (14), the result is

$$C_W = - \int_0^{2\pi} \left[ \bar{C}_{M_M} + \bar{C}_{M_{UR}} \cos \omega t - \bar{C}_{M_{UI}} \sin \omega t \right] \bar{\alpha} \sin \omega t \; d(\omega t) \qquad (17)$$

(The integration range, $0 \le \omega \le 2\pi$, is equivalent to one complete cycle of motion.) After the integrations indicated in Eq. (17) are performed, it is found that the term involving the mean moment vanishes as well as the term containing the real part of the unsteady moment; the final result for the theoretical work coefficient is given by

$$C_W = \pi \, \bar{\alpha} \, \bar{C}_{M_{UI}} \qquad (18)$$

This is the work done by the air on the airfoil; hence, a positive value of $C_W$ will indicate an unstable motion, since this implies a net energy exchange from the surrounding medium to the airfoil, whereas a negative value of $C_W$ will indicate a stable, or damped motion.

Carta, F. O., & Niebanck, C. F. (1969). *Prediction of rotor instability at high forward speeds.* (Vol. III, Stall).

I draw your attention to formula 18 where the unsteady component is responsible for the work done in a single degree of freedom system. This means that the phase shift between the angle of attack and the moment leads to aerodynamic damping.

The following paper dives into this topic in greater detail. This and another paper were provided as citations in the offending sentence to give this very context.

Bowles, P. O., Corke, T. C., Coleman, D. G., Thomas, F. O., & Wasikowski, M. (2014). Improved Understanding of Aerodynamic Damping Through the Hilbert Transform. *AIAA Journal*, *52*(11), 2384–2394. https://doi.org/10.2514/1.J052630

I find the description in terms of $\frac{dC_l}{d\alpha}$ or $\frac{dC_m}{d\alpha}$ less succinct. The phase difference description is applicable to different modes of instability.

**Cycle to Cycle Variations.**

While in the long run the system may oscillate around an attractor it can have large departures from that attractor in the short run, do we still want to call that cycle stable? The work done formula presented on the previous page presents the boundary integral over a cycle, not over a long run range of cycles, so stability is determined cycle to cycle and can even be determined with in the cycle referencing Bowles and Corke again.

**Towing Tank Outlier Case**

The towing tank case presents data on a single pressure port because as you said, we didn't have full lift available. In the limited part of the aerodynamic data that I had available, there weren't any good examples to demonstrate the outlier detection.

**Figure 22**

There is only a small different between the clusters this was already highlighted but I have extended the comment. The misplaced reference was fixed.

**Figure 12**

The colors represent z-scores- it's standardized, added comment to caption.

**Figure 14-16**

The probability distributions are presented with minimum possible detail.

**Section 4.1**

I agree. I have moved it and I believe it reads better now.

**Page 15 Complicated Case**

From the limited set of data available to me, this demonstrated the effects best.

**Unnecessary figures.**

I removed some figures.

**Referee #2 Tuhfe Gocmen**

Section 3

1. Fixed
2. Moved to later but final placement will be done by the editors.
3. The clustering algorithm receives a time series as input.
4. A full study about the downsampling is published in my students thesis I have highlighted this.

Section 4.

- Agreed and rearranged.
- Regarding the CNN architecture. I am in a split mind. I used the generic pictures to try and educate the reader in broad terms.
- You are right about the uncertainties on the training data. This is why I said I assume the training data is correct. I did a few trials seeing how different my clicks were on the same figure, it was less than the difference in the convection speeds by a good margin. It's not a perfectly satisfying answer, but the best approach I could think of. The approach is intended to be used to analyse experimental data. If it was an online system, I would definitely want to be more rigorous.
- I added some explanations of the training process.

Regarding my suggestions for future unsteady aerodynamics models, I was talking in the sense of future work and where ML could be applied, it would be a different model completely however.

The code example will have some of the dataset.

[revised manuscript text omitted]

---

## Author Response (AR2)

A general comment regarding formatting. The journal reformats and typesets the paper into a two column mode. All of my figures have been designed to fit efficiently in this mode but I have to submit in the draft phase as single column. In short, the paper will visually appear a lot better after typesetting. Nonetheless, I forced the figures to be nearer to their reference in this submission.

Change List Report 1:

Thank you for the comments, they were helpful.

- The stall is bad comment is removed.

- Google URLS Removed

- Citation Date fixed.

- Strangfeld Corrected

- [33] The effective decambering analogy doesn't take into account the increase in drag.

- [41] List item changed to match suggestion, a nice neat sentence now.

- Described conditions of deep stall initiating from the leading edge, more accurate now.

- spike and dump removed.

- [160] I understand the objection. However, the towing tank provides a very good example for outlier detection as we had access to raw data. The Muller-Vahl dataset has been fairly well sanitized with most data with experimental issues removed from the dataset. For this reason it was unfortunately necessary to make things a bit more confusing in the paper by including the two datasets.

- [223] Figure 6 now referenced. Spelling fixed. The descriptive equations can be found in the reference.

- [figure 9.] I have added some sentences explaining where each of the datasets is used.

- [340] I refined my explanation. If I was to describe the methods and their motivation and then pour through the full dataset, it would require a whole book. I use enough cases to demonstrate the variability of dynamic stall and the methods used. That is the aim of the paper.

Change List Report 2:

Thank you for the detailed response. The comments have really helped me refine the paper.

- The stall is bad comment is removed.

- Regarding the training method. I concede your point regarding the lack of low fidelity models and the poor performance of 2D CFD.

I have included some comments regarding this. And yes the variant that would take into account VIV would have to be a 3D model, this is also possible with this approach.

- Stability: I have changed some of the language.

- Added the missing reference to figure 9.

- Probability Distributions - I deliberately only included a minimum of information. I wanted to show that there is a change in mean and variance and not any deeper details.

I believe that now all outstanding issues have been resolved. I also re-proof read the paper.

---

## Author Response (AR3)

Thanks for the detailed comments. They were very helpful and helped me address the concerns of the previous reviewers a bit better. I know it's a tough read. It's been a very tough topic for me to communicate the ideas properly even in person. There are some subtle ideas in machine learning that have large implications for us in wind energy but I would need a paper double the length to make the point in full. I hope this manifestation of the paper now works.

Overall:

The paper overall seems to be missing a bit of a roadmap (both overall and within individual sections). It reads now as somewhat ad hoc and taking a trial and error approach rather than being systematic and purposeful in the approach. The overall justification and motivation for the whole study seems buried in the conclusions and should be brought front and center. See my notes on lines 443 to 452.

Through out the paper. I have added significantly more signposting and tried to explain the context of the paper in a few more places. In addition to the specific comments here. I re-read the entire paper and added many clarifications. While the content hasn't changed, I feel that this version should do a much better job of explaining the content.

Introduction section 1

In the first section, the discussion and overview of stall and the literature is quite well done as is the introduction to the machine learning methodology. However, the paper then abruptly transitions into the data, methods and analysis. Given this, it is not surprising the reviewers critiqued the lack of explanation around choice of data selection, confusion about selection of model fidelities, etc… the rest of the paper following section 1 seems to come out of nowhere. Consider adding a subsection and/or paragraph at the end of section 1 that:

- At a high level describes what you will do in the rest of the paper - what do you plan to do in a concrete way?

- Provides basic rationale for taking this approach (one cannot do everything in a single paper so it is okay to say that you are choosing x,y,z approach as an illustration of a more generalizable approach, etc)

- Gives an overview of what the rest of the paper will do section by section

There is a new paragraph in the first section that lists the aims of the paper i.e. each of the models and also provides context that the paper is aiming to demonstrate machine learning methods as a viable tool.

Experimental Section 2

Line 15-19 – paper jumps into technical details without proper motivation. Why do we care? See comment on line 24 – how do we use these models? What is meant by viable? Why do we want the models to be better?

I added an extra sentence to this paragraph stating : and this matters for designing wind turbines and experimental data. Viable has been clarified. Justification of better models

Line 20 - In text mention of Stangfeld still incorrect (should be Strangfeld)

I am being haunted by LaTex. It should be fixed now.  Really….

> Müller-Vahl, H., Strangfeld, C., Nayeri, C. N., and Paschereit, C. O.: Control of Thick Airfoil, Deep Dynamic Stall Using Steady Blowing, AIAA Journal, 53, 1–34, http://arc.aiaa.org/doi/abs/10.2514/1.J053090, 2015.

Line 24 – what do you mean by viable? Computationally efficient? Convergent?

Clarified.

Line 26 – for terms, do you mean definitions used within the literature for stall?

Modified the sentence

Line 92 – incomplete sentence after semicolon

Fixed

Line 128-129 – sentence incomplete

I could not find what you meant, I re-read this paragraph a dozen times.

Line 149-150 – syntax error in sentence

Fixed

Lines 156-159 – introduction to experimental data is abrupt and weak – why this data? Can you motivate its selection somehow?

I introduced the data section.

Experimental Section 3

General comments: Again a small introduction here that lays the roadmap of the section and ties the elements together upfront (here is what im going to tell you) would help. Generally, the section is very dense with text and uses a lot of terminology that is unfamiliar to the wind domain. A few more images could go a long way. Figure 8 is a good start but more could be done.

New paragraph added for context. I went through the sections and tried to add more plain language sentences to clarify in between. I haven't added figures, I hope that the additional explanations is enough. I am resisting on this point because my paper is already quite long and the costs are accumulating rapidly.

Line 245 – behavior

Changed to behaviour.

Line 262-266 – important context, would be worth bringing up front a bit more and repeating here as necessary

There was a mention of this point albeit briefly. I added more detail to emphasis the point.

*Treating stall as a stochastic process is a relatively recent idea. As early as 1978, one sees acknowledgment that stall is variable in literature such as \citet{Stall}; an experimental report that described it taking measurements of 50 cycles of a pitching airfoil undergoing dynamic stall to ensure convergence of the lift.* **While these researchers did acknowledge the variability of the data, they still used a simple average to represent the data. This was a reasonable choice at the time given that many of the more advanced tools now available did not exist nor was the requisite computational power available.** *Only more recently have researchers have begun to address the spatial and temporal variability of stall in experimental work. \citet{Mulleners2013} were able to show that dynamic stall could be described by two stages of a shear layer instability, and that the development of these instabilities varied across cycles. In light stall, it was shown that the trailing edge separation region had two modes, resulting in either a Von Karman shedding pattern or a stable dead water zone \citep{Mulleners2016}. The separation pattern fluctuates unreliably and when vorticity is present, the vortex convection speed is also variable.*

Results Section 4

Consider adding a paragraph in front of section 4.1 to motivate section 4 overall – right now it seems to be a presentation of results without context or critical analysis. It reads more as a demo of the methods. Perhaps results is the wrong title for this section

Yes it was intended more of a demonstration of the methods. I have emphasised this and renamed the section. Examples.

Line 327 – can you be more descriptive at all about a number of test configurations (describe them before jumping into them).

I have added some extra sentences and split the two cases. I have made the lessons learned from each case a bit clearer hopefully

Section 4.1 – consider elaborating on purpose: what are you achieving here? What is the purpose of demonstrating the patterns?

Added some lesson learnt sentences: tldr: Averages suck, there are better ways of extracting information from data.

Section 4.2 – motivate stronger

Added arguments here. Almost labouring the point.

Line 341 / Line 368 – strengthen these sentences significantly. Why does this matter? Give more concrete justification and or application context

Section 5

Line 370 – consider qualifying outliers… outliers that may corrupt the quality of the dataset. Add For instance in front of wind tunnels to explicitly link the sentences together

Changed

Line 381 – 384 – this paragraph seems to be key to why this paper matters… can you speak more to this both up front and in the results and analysis sections?

Added two sentences to emphasis this point.

Second half of section 5 – it gets a bit lost between what here is new work versus what you are extracting from the thesis of Steenbuck. The section also ends abruptly.

I added paragraph at the end and changed the reference to steenbucks thesis.

Section 6 Conclusions

Line 428 – syntax error, missing "to"

Fixed

Context in 416-444 is the type of material that should be in the introduction and not buried in the conclusions

Consider eliminating lines 443 to 452. Providing a specific recipe for a multi-fidelity machine-learning driven approach to modeling aerodynamics accounting for stall has pitfalls that I think the reviewers have already highlighted. Just speaking to the fact that machine learning driven models, like the ones presented in this paper, can be used in tandem with models of varying levels of fidelity and data sets to build robust and computationally unsteady aerodynamic models is feasible. This paper presents first steps towards enabling such an approach by applying machine learning techniques to learn complex and dynamic stall behavior from a large set of test data.

I haved changed the conclusion so that the new potential model is in it's on section. I retained the exact recipe because it still makes an important point about hierarchies of data. I have highlighted this explicitly in the text now.

---

## Author Response (AR4)

Thanks for the comments. Paragraph 2 of the paper now does the signposting in addition to the previous signposting.

The logic is now.

Airfoil stall is complicated to model, making wind turbine development hard.

Machine learning can help, we will demonstrate how.